# Ablation of prostaglandin E$_2$ signalling through dual receptor knockout in CAR T cells enhances therapeutic efficacy in solid tumours

Janina Dörr [1,15], Lisa Gregor [1,15], Sebastian B. Lacher[2,3,15], Arman Oner[1], Yi Sun[4,5], Ignazio Piseddu[1,6,7], Luisa Fertig[1], Sebastijan Spajic[1], Stefanie Lesch[1], Stefanos Michaelides[1], Matthias Seifert [1], Adrian Gottschlich [1,8], Natasha Samson[1], Lina Majed[1], Daria Briukhovetska [1], Donjetë Simnica[1], Viktoria Hartmann[1], Kathrin Gabriel [1], Sonia Cohen[9], Genevieve M. Boland[5,9], David Andreu-Sanz[1], Emanuele Carlini[1], Sophia Stock [1,8,10], Anne Holtermann[1,10], Philipp Jie Müller[1], Thaddäus Strzalkowski[1], Marcel P. Trefny [1], Stefan Endres[1,10,11], Russell W. Jenkins [4,5], Jan P. Böttcher[2,12,13,16] & Sebastian Kobold [1,10,11,14,16] ✉

The efficacy of chimeric antigen receptor (CAR) T cell therapy in solid cancers is limited by immunosuppression in the tumour microenvironment (TME). Prostaglandin E$_2$ (PGE$_2$) is a key factor locally inhibiting T cell function. We hypothesized that targeted ablation of PGE$_2$ signalling in CAR T cells may enhance their activity in PGE$_2$-rich solid tumours. Here we generate knockout CAR T cells double deficient for the PGE$_2$ receptors EP2 and EP4 (EP2$^{-/-}$EP4$^{-/-}$) by CRISPR–Cas9 engineering. EP2$^{-/-}$EP4$^{-/-}$ CAR T cells expanded unabatedly in the presence of PGE$_2$. Further, they effectively controlled syngeneic and human xenograft tumour models in vivo, which was accompanied by intratumoural accumulation and persistence of modified T cells. Improved anti-tumour activity was also observed against patient-derived tumour samples from patients with pancreatic ductal adenocarcinoma (PDAC), colorectal (CRC) and neuroendocrine (NET) cancer. Our data uncovers the detrimental impact of PGE$_2$-mediated suppression on CAR T cell efficacy and highlights EP2 and EP4 targeting as a potential strategy.

Chimeric antigen receptor (CAR) T cells have achieved remarkable response rates in otherwise refractory haematologic malignancies and led to the first approval by the US Food and Drug Administration (FDA) in 2017 (ref. 1). A CAR is a synthetic protein introduced into T cells, which is composed of an extracellular single-chain variable fragment targeting a specific tumour surface antigen, a hinge/spacer and transmembrane domain, an intracellular co-stimulatory domain and an intracellular signalling domain triggering T cell activation[2,3].

The advantage of CAR T cells lies in their ability to merge the specificity of an antibody with the cytotoxic effect of activated T cells in a human leukocyte antigen (HLA)-independent manner[2,3]. In some patients, a single infusion was sufficient to cure refractory disease[4–6]. For many years, clinical development has focused on haematological malignancies, but efforts are increasingly expanded to solid tumour entities, such that the clinical trial landscape is growing rapidly[7]. This development is certainly propelled by the success of immune checkpoint

**Fig. 1 | Knockout of Ep2 and Ep4 improves the efficacy of adoptively transferred OT-I T cells by increasing their persistence in the TME. a**, $5 \times 10^6$ CD45.1 × *CD4$^{Cre}$Ptger2$^{-/-}$Ptger4$^{fl/fl}$* OT-I T cells and $5 \times 10^6$ CD90.1 OT-I T cells were co-injected into D4M.3A-SIINFEKL ($10^6$ cells s.c.) bearing mice. **b,c**, On days 2, 5, 9 and 14, the abundance of T cell populations in tumours (**b**) and lymph nodes (**c**) was determined by flow cytometry (data shown as mean ± s.e.m. of $n = 5$ mice per group). Statistical analysis was done with a two-way ANOVA. **d**, OT-I T cells with Ep2 and/or Ep4 knockout were treated with 2,000 ng ml$^{-1}$ PGE$_2$ and cAMP levels were determined in a luciferase-based readout (data shown as mean ± s.d. of $n = 3$ independent experiments with 2 technical replicates each; statistical analysis was done with an ordinary one-way ANOVA). **e**, OT-I T cells with Ep2 and/or Ep4 knockout were treated with 1,600 ng ml$^{-1}$

PGE$_2$ and CREB phosphorylation was measured by flow cytometry. Pooled data of $n = 3$ representative experiments (mean ± s.d.) with 2 technical replicates each and a representative change of the pCREB MFI are shown. Statistical analysis was done with a two-way ANOVA. Mice bearing D4M.3A-SIINFEKL tumours ($10^6$ cells s.c.) were treated with $10^7$ adoptively transferred (i.v.) OT-I T cells with Ep2 and/or Ep4 knockout. **f,g**, Tumour growth (**f**) was monitored over time and survival (**g**) was determined. Pooled results of $n = 3$ independent repetitions with $n = 5$ mice per group are shown as mean ± s.e.m. Statistical analysis was done with a repeated measurements mixed-effects analysis with Dunnett's multiple comparison correction (**f**) and log-rank (Mantel–Cox) test (**g**). Panel **a** created with BioRender.com.

inhibition in many solid tumours, which demonstrates that T cells can elicit potent anti-tumour effects, mainly in patients with pre-existing immunity and active T cells[8]. Notable antigens in these trials include human epidermal growth factor receptor 2 (HER2)[9] or the epidermal growth factor receptor and its variant (EGFR/EGFRvIII)[10,11]. Despite limited success in solid cancer patients thus far, there is emerging evidence that engineered T cells, and specifically CAR T cells, may enhance therapeutic responses[12–14].

Currently, major limitations of CAR T cell therapy in solid tumours include lack of T cell infiltration into the tumour[15,16], T cell exhaustion and functional suppression[15], as well as limited persistence and expansion of CAR T cells in the tumour microenvironment (TME)[2,3]. A well-described immunosuppressive molecule in the TME is the endogenously produced bioactive lipid prostaglandin E$_2$ (PGE$_2$)[17–19]. Elevated

levels of PGE$_2$ and its rate-limiting enzyme, COX-2, are associated with poor prognosis and reduced survival in various cancer types, such as breast, prostate or pancreatic cancer[20–22]. While PGE$_2$ does have direct pro-tumour effects, mainly through antiapoptotic signalling[23] or promotion of tumorigenic angiogenesis[24], recent evidence suggests that PGE$_2$ mainly contributes to tumour progression through inducing immunosuppressive effects[18,19,25–27].

While a preventive effect of COX inhibitors on cancer progression has been observed for various types of cancer[20,28–30], therapeutic use has failed to demonstrate a clear benefit of COX inhibition in clinical trials[25,28,30–33]. A major issue related to systemic inhibition of COX-dependent PGE$_2$ lies in its homeostatic functions, which results in treatment-limiting side effects that narrow the therapeutic window[31–36]. This emphasizes the need for alternative and more tailored methods

to target the COX-2–PGE$_2$ axis, restricting its effects to a well-defined effector cell population.

Extracellular PGE$_2$ signals via PGE$_2$ receptors (EP), namely EP1, EP2, EP3 and EP4, all of which belong to the family of G protein-coupled receptors (GPCRs)[17]. These receptors are expressed on a variety of cell types that have important roles in shaping the TME, including myeloid-derived suppressor cells, macrophages, dendritic cells, natural killer cells and T cells[18,19,25–27,37]. We and others have recently uncovered that PGE$_2$ mediates tumour immune escape through its receptors EP2 and EP4 by hindering the differentiation and expansion of tumour-infiltrating anti-cancer T cells through disruption of interleukin (IL)-2 signalling[38–41]. This raises the fundamental question of whether safeguarding T cells from PGE$_2$ can enhance their functionality within solid tumours and thereby facilitate tumour eradication. Thus, we sought to investigate whether abrogating the effects of tumour-derived PGE$_2$ on therapeutic T cells can increase their functionality within solid tumours and enhance the efficacy of T cell cancer therapy.

We thus developed a clustered regularly interspaced short palindromic repeats (CRISPR)–CRISPR-associated protein 9 (Cas9)-based knockout system for primary T cells[15] to eliminate the EP2 and EP4 receptors, as well as their subsequent inhibitory signalling, selectively in CAR T cells. Given the therapeutic challenges associated with targeting GPCRs such as EP2 and EP4, we aimed to establish novel functional readouts to assess gene editing efficiency. Furthermore, we demonstrate how the concurrent elimination of these two receptors enhances CAR T cells' ability to proliferate and, as a result, increases their therapeutic potential, both in vitro and in vivo in syngeneic as well as xenograft mouse models of pancreatic carcinoma, melanoma and mesothelioma and against different patient-derived organotypic tumour spheroids (PDOTS).

## Results

### Knockout of EP2 and EP4 improves the efficacy of adoptively transferred OT-I T cells by increasing their persistence in the TME

As demonstrated recently, PGE$_2$ has detrimental effects on the establishment of an efficient T cell response against solid tumours[21,42]. Shielding T cells from tumour-derived PGE$_2$ by knocking out its receptors EP2 and EP4 on therapeutically transferred T cells therefore might be a promising strategy to improve adoptive T cell transfer such as CAR T cell therapy. To verify whether EP2 and EP4 double knockout can rescue the function of transferred T cells comparably to what was reported recently, we performed in vivo tracking of adoptively transferred ovalbumin (OVA)-specific OT-I T cells using congenic marker-positive T cells in a murine D4M.3A-SIINFEKL tumour model. We co-injected equal numbers of T cells obtained from a CD90.1 wild type and a CD45.1 × *CD4$^{Cre}$Ptger2$^{-/-}$Ptger4$^{fl/fl}$* mouse, both transduced with the OT-I receptor at equal efficiencies (Fig. 1a). Flow cytometry analysis (Extended Data Fig. 1a) of tumours, lymph nodes, spleens and blood after 2, 5, 9 and 14 days revealed higher numbers of *CD4$^{Cre}$Ptger2$^{-/-}$Ptger4$^{fl/fl}$* T cells in tumours from day 5 onwards (Fig. 1b). However, no consistent differences were observed regarding T cell numbers between wild-type and *CD4$^{Cre}$Ptger2$^{-/-}$Ptger4$^{fl/fl}$* T cells in lymph nodes (Fig. 1c), spleen (Extended Data Fig. 1b) and blood (Extended Data Fig. 1c), confirming that the changes in composition are specific to the PGE$_2$-rich TME. Together, these data indicate that PGE$_2$ in the TME severely undermines effectivity of adoptively transferred T cells by impairing their local persistence. Along these lines, when transducing *CD4$^{Cre}$Ptger2$^{-/-}$Ptger4$^{fl/fl}$*-derived splenocytes with an anti-epithelial cell adhesion molecule (EpCAM)-CAR, we could observe improved tumour control and clearance in 2/5 mice in a subcutaneous murine Panc02-OVA-EpCAM model (Extended Data Fig. 1d). Together, these data indicate that adoptively transferred T cells benefit from a knockout of the PGE$_2$ receptors EP2 and EP4, testified by their improved intratumoural accumulation and therapeutic activity.

To assess the therapeutic potential of these findings, we developed a CRISPR–Cas9 knockout method to disable Ep2 and Ep4 signalling in tumour-targeting T cells. As a first proof of concept, we introduced both Ep2 and Ep4 single knockouts, as well as a simultaneous knockout of both receptors into OT-I T cells (Extended Data Fig. 1e). Importantly, we were unable to establish specific and reliable Ep2 or Ep4 detection by flow cytometry or western blot, a common issue with GPCRs, which complicates knockout validation as protein-level reduction cannot be directly measured to assess gRNA efficacy or CRISPR optimization. This, until now, has hindered many GPCRs from being evaluated regarding their function in a therapeutic setting. To overcome this key limitation, we herein propose evaluating intractable knockout targets by monitoring reductions in their downstream signalling. To this end, we established assays measuring the production of cyclic adenosine monophosphate (cAMP) and downstream phosphorylation of cAMP-response element-binding protein (CREB) upon PGE$_2$ stimulation as a proxy for a reduction in the expression of the PGE$_2$ receptors Ep2 and Ep4. Upon double knockout of both receptors, downstream signalling following PGE$_2$ stimulation, measured by intracellular cAMP levels (Fig. 1d) and CREB phosphorylation (Fig. 1e and Extended Data Fig. 1f), was completely abrogated. By contrast, the individual knockout of either Ep2 or Ep4 was not sufficient to prevent PGE$_2$ signalling in OT-I T cells.

We previously observed improved tumour control when using *CD4$^{Cre}$Ptger2$^{-/-}$Ptger4$^{fl/fl}$* TCF1$^+$ stem-like CD8$^+$ OT-I T cells compared with wild-type OT-I T cells in adoptive cell transfer experiments. To probe whether engineered and differentiated cells would phenocopy this effect, we chose to treat mice bearing established D4M.3A-SIINFEKL melanomas with these cells. Ep2$^{-/-}$Ep4$^{-/-}$ but not Ep2$^{-/-}$ nor Ep4$^{-/-}$ OT-I T cells were able to slow tumour growth (Fig. 1f and Extended Data Fig. 1g). Ep2$^{-/-}$Ep4$^{-/-}$ but also Ep4$^{-/-}$ OT-I T cells prolonged survival (Fig. 1g) compared with mock OT-I T cells (electroporation without ribonucleoprotein (RNP) complex), whereby the effect of Ep2$^{-/-}$Ep4$^{-/-}$ was superior to that of Ep4$^{-/-}$ OT-I T cells. Remarkably, we observed complete responses in 3 out of 15 mice in the Ep2$^{-/-}$Ep4$^{-/-}$ and in 1 out of 15 mice in the EP2$^{-/-}$ groups, while no complete responses occurred in mice treated with mock CRISPR or Ep4$^{-/-}$ T cells. Consistent with our hypothesis, shielding tumour-targeting T cells by knocking out Ep2 and Ep4 improved the effectiveness of adoptive cellular therapy in the presence of PGE$_2$ and constitutes a promising therapeutic avenue.

### Generation of CAR T cells with a CRISPR-based knockout of EP2 and EP4

To transfer these findings into a therapeutic CAR T cell protocol, we next integrated the CRISPR–Cas9-based knockout of EP2 and EP4 into our retroviral transduction protocol to produce murine and human CAR T cells targeting the murine tumour antigen EpCAM or the human tumour antigens HER2, HER1 or mesothelin (MSLN) (Fig. 2a). We routinely obtained around 50–80% CAR-transduced T cells (Extended Data Fig. 2a), which was comparable between the knockout and mock CRISPR conditions. Equivalent to our OT-I data, we observed a complete shutdown of the PGE$_2$ signalling pathway, as measured by cAMP levels (Fig. 2b) and CREB phosphorylation (Fig. 2c and Extended Data Fig. 2b) in EP2$^{-/-}$EP4$^{-/-}$ CAR T cells but not in EP2$^{-/-}$ or EP4$^{-/-}$ CAR T cells upon stimulation with PGE$_2$. Of note, unphysiologically high concentrations of PGE$_2$ (murine setting 1,600 ng ml$^{-1}$ and human setting 2,000 ng ml$^{-1}$) were purposefully chosen to probe the robustness of the receptor knockouts. To access knockout efficiency quantitatively for our therapeutically relevant knockout in human CAR T cells, we subsequently performed MiSeq Illumina sequencing of human anti-HER2 CAR T cells with the respective EP2 and EP4 knockouts, serving as a validation of our previously described functional readout of knockout efficiency. We obtained efficiencies of on average 70% for EP2 and 90% for EP4 receptor both in the single and double knockout settings, proving high knockout

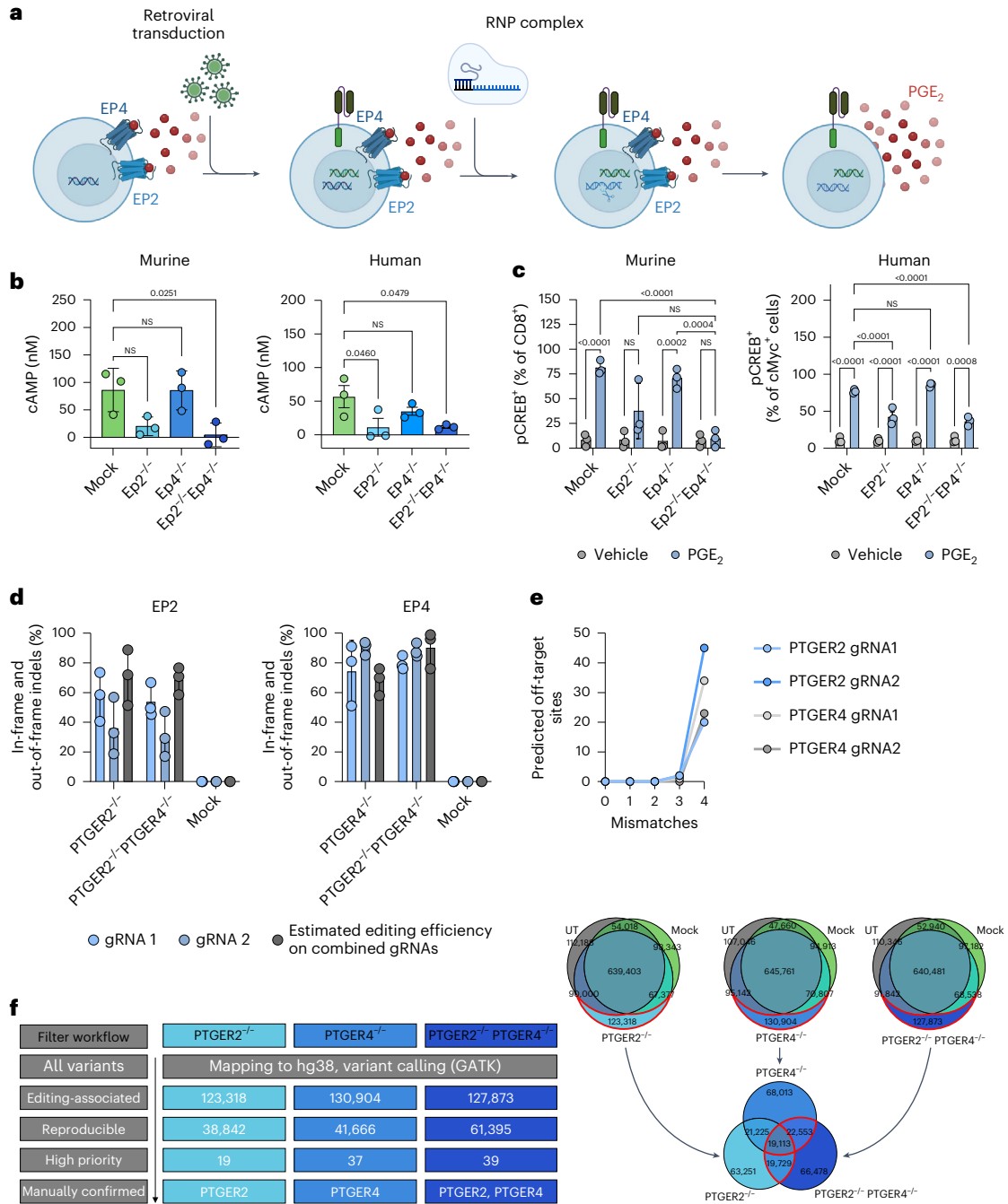

**Fig. 2 | Generation of CAR T cells with a CRISPR-based knockout of EP2 and EP4. a**, Primary murine and human T cells were retrovirally transduced to express a second-generation CAR targeting EpCAM or HER2, respectively. After 24 h, CAR T cells were submitted to CRISPR-based gene editing to knockout the surface receptors EP2 and EP4. **b**, Murine and human CAR T cells with EP2 and/or EP4 knockout were treated with 2,000 ng ml$^{-1}$ PGE$_2$, and cAMP levels were determined in a luciferase-based readout (data shown as mean ± s.d. of $n$ = 3 independent experiments with 2 technical replicates each; statistical analysis was done with an ordinary one-way ANOVA. **c**, CAR T cells with EP2 and/or EP4 knockout were treated with 1,600 ng ml$^{-1}$ (murine) and 2,000 ng ml$^{-1}$ (human) PGE$_2$. CREB phosphorylation was measured by flow cytometry. Pooled data of $n$ = 3

representative experiments with 2 technical replicates each (mean ± s.d.) and a representative change of the pCREB MFI are shown. Statistical analysis was done with a two-way ANOVA. **d**, Genomic DNA samples from human anti-HER2 CAR T cells of $n$ = 3 donors with their respective EP2 and EP4 knockouts were submitted to MiSeq Illumina sequencing. Data are shown as mean ± s.d. Statistical analysis was done with a two-way ANOVA. **e**, Potential off-targets were predicted for each gRNA using Cas-OFFinder allowing up to four mismatches between gRNA and DNA sequences. Depicted are the numbers of predicted off-targets per gRNA. **f**, Workflow for variant filtering during unbiased variant identification from whole genome sequencing data. Panel **a** created with BioRender.com.

efficiency (Fig. 2d and Extended Data Fig. 2c,d). Together, we are the first to engineer a double knockout of two unstainable receptors, namely EP2 and EP4, and demonstrate feasibility of knockout integration into a retroviral CAR T cell manufacturing workflow with high efficiency.

To further characterize our human EP2$^{-/-}$EP4$^{-/-}$ CAR T cell knockout product, we also assessed potential off-targets. To this end, we in silico predicted off-targets for all our gRNAs using Cas-OFFinder[43]. We included all predicted off-target sites with up to 4 mismatches between gRNA and DNA sequences in our analysis, as 97% of off-targets are

anticipated to have less than 5 mismatches between gRNA and DNA sequences[44]. No off-targets with 0–2 mismatches were predicted for all gRNAs. Up to 2 off-targets with 3 mismatches per gRNA and up to 45 off-targets with 4 mismatches were predicted (Fig. 2e). To validate those predicted potential off-targets, we preformed whole genome sequencing for one human donor. All Cas-OFFinder predicted sites were manually inspected in the integrative genomics viewer (IGV), and no signs of variations compared with our unmodified control samples could be found. As there are also gRNA-independent off-target effects[45], we also performed genome analysis toolkit (GATK) variant calling in our samples. We excluded all variants from the analysis that were also present in our unmodified and mock CRISPRed samples, as those are likely normal variations from our donor to the reference genome that are unrelated to our engineering approach. We further excluded all variants that were not reproducible between the EP2 and EP4 double knockout and the respective single knockouts, as we deemed them of unlikely relevance for lack of reproducibility within a single donor. From the remaining variants, we selected only those that are likely to have an influence on a protein (high-priority variants; Fig. 2f). We defined a high-priority variant as a variant located in the exonic or splicing region with a frequency of less than 0.01 in the 1000 Genomes database and a deleterious prediction from at least one of SIFT, Polyphen, MutationTaster and CADD algorithms. The remaining variants of interest (Supplementary Table 1) were again manually inspected in the IGV, and the only confirmed variants were PTGER2 and PTGER4 in their respective knockout samples, representing good on-target efficiency and validity of our approach for variant identification. Taken together, we were neither able to confirm any predicted off-targets nor independently find any off-target sites in our donor, pointing towards a potential good safety profile of our knockout strategy.

### Improved expansion and survival of EP2$^{-/-}$EP4$^{-/-}$ CAR T cells in the presence of PGE$_2$ leads to enhanced anti-tumour function in vitro

Next, we evaluated the performance of PGE$_2$ receptor knockout CAR T cells in vitro. While pre-incubation of mock CAR T cells with PGE$_2$ over 48 h (Extended Data Fig. 3a) completely abrogated their killing capacity of murine pancreatic ductal adenocarcinoma (PDAC) Panc02 cells overexpressing EpCAM (Fig. 3a), we observed a partial rescue of tumour control for Ep2$^{-/-}$ and Ep4$^{-/-}$ CAR T cells. However, only Ep2$^{-/-}$Ep4$^{-/-}$ CAR T cells were able to completely eradicate tumour cells in the presence of PGE$_2$ (Fig. 3a,b and Extended Data Fig. 3b). We made similar observations in a human setting, whereby only EP2$^{-/-}$EP4$^{-/-}$ anti-HER2 CAR T cells were able to control growth of BxPC3 tumour cells (Fig. 3c,d and Extended Data Fig. 3b,c). As the pre-incubation setting (Extended Data Fig. 3a) allows for PGE$_2$-treatment-induced changes of T cell numbers and previous reports show deficient T cell

proliferation in the presence of PGE$_2$ (refs. 42,46,47), we next evaluated whether the observed differences in killing capacity are due to differential T cell activation and cytotoxicity or due to changes in T cell numbers. Indeed, we observed decreased proliferation, as well as impaired survival of anti-EpCAM CAR T cells (Fig. 3e,f) and anti-HER2 CAR T cells (Fig. 3g) in the presence of PGE$_2$. This phenotype could be completely rescued by knocking out EP2 and EP4. We did not see an inhibitory effect of PGE$_2$ on T cell activation, as interferon (IFN)-γ production at the single-cell level of murine anti-EpCAM CAR T cells was unabated regardless of PGE$_2$ (Fig. 3h). Similarly, no changes in CD25 nor CD69 expression on human anti-HER2 CAR T cells were observed (Fig. 3i,j) when stimulating cells with their respective HER2 tumour antigen in vitro. This suggests that failure to eradicate tumour cells in the presence of PGE$_2$ results from impaired CAR T cell proliferation but not inhibition of their killing capacity. These observations are consistent with in vivo results presented in Fig. 1a,b. Taken together, EP2 and EP4 double-knockout CAR T cells show improved functionality in the presence of PGE$_2$ in vitro.

### EP2 and EP4 knockout enhances anti-tumour activity of CAR T cells in vivo and in patient-derived samples

We further evaluated the in vivo performance of PGE$_2$ receptor knockout HER2-targeting CAR T cells in a xenograft model derived from BxPC3 PDAC cells. EP4$^{-/-}$ as well as EP2$^{-/-}$EP4$^{-/-}$ anti-HER2 CAR T cells reduced tumour growth and consequently prolonged survival, while EP2$^{-/-}$EP4$^{-/-}$ CAR T cells were able to clear the tumours of three mice (Fig. 4a,b). To further validate the functionality of our receptor knockouts in a second mouse model, we implemented a mesothelioma model using the Msto-hMSLN cell line in NOD-Prkdcscid-IL2rgTm1/Rj (NXG) mice, which were treated with $2 \times 10^5$ EP2$^{-/-}$EP4$^{-/-}$ anti-MSLN CAR T cells. Mice showed reduced tumour growth, resulting in prolonged survival compared with mice treated with mock electroporated cells, with three mice transiently clearing their tumour (Fig. 4c,d). Building on these promising preclinical findings, we sought to evaluate the therapeutic potential of our receptor knockout CAR T cells in a clinically relevant setting. To this end, we tested EP2$^{-/-}$EP4$^{-/-}$ anti-HER1 CAR T cells in patient-derived tumour explants from colorectal carcinoma (CRC), PDAC and neuroendocrine tumours (NET). PDOTS were co-cultured in a three-dimensional (3D) microfluidic culture device with therapeutic CAR T cells, after which tumour cell viability was assessed through fluorescent nuclear staining and subsequent imaging (Fig. 4e). Increased anti-tumour activity of EP2$^{-/-}$EP4$^{-/-}$ anti-HER1 CAR T cells was observed against three different CRC patient samples (Fig. 4f) compared with untransduced (UT) T cells indicated by a decrease in tumour cell viability. Double-knockout CAR T cells further showed an increased benefit against samples from PDAC (Fig. 4g) and NET (Fig. 4h). As anti-HER1 CAR T cells showed no effect against gastrointestinal stromal tumour samples (Fig. 4i), a tumour known

---

**Fig. 3 | Improved expansion and survival of EP2$^{-/-}$EP4$^{-/-}$ CAR T cells in the presence of PGE$_2$ leads to enhanced anti-tumour function in vitro.**
**a**–**d**, Murine anti-EpCAM CAR T cells (**a**,**b**) and human anti-HER2 CAR T cells (**c**,**d**) were pretreated with 250 ng ml$^{-1}$ (murine) or 500 ng ml$^{-1}$ (human) PGE$_2$ for 48 h, after which co-culture experiments with Panc02-OVA-EpCAM or BxPC3 cells were performed. Target killing was measured using impedance-based readout. A representative experiment (**a**,**b**) of $n = 3$ independent repetitions with 3 technical replicates each is shown as mean ± s.d., (**b**,**d**) as well as the area under the curve (AUC) as a proxy for the quantification of the tumour cell lysis over all repetitions. AUC data are shown as mean ± s.d. (**b**,**d**). Statistical analysis was done using a two-way ANOVA with Šidák's multiple comparison correction. **e**, To assess their proliferative capacity, murine anti-EpCAM CAR T cells were cultured with 250 ng ml$^{-1}$ PGE$_2$ for 24 h and 48 h, after which live cell numbers were determined using counting beads. **f**, Viability of the murine anti-EpCAM-CAR T cells was assessed using flow cytometry after 24 h and 48 h. **g**, To assess their proliferative capacity, human anti-HER2 CAR T cells were cultured with 500 ng ml$^{-1}$ PGE$_2$.

Cell numbers were determined using counting beads. Results are shown as normalized ratios calculated from treated to untreated values depicted as mean ± s.d. for $n = 3$ independent experiments with 3 technical replicates each (**e**–**g**). Statistical analysis was done with a repeated measurements two-way ANOVA with Dunnett's multiple comparison correction (**e**,**f**) or ordinary one-way ANOVA with Tukey's multiple comparison correction (**g**). **h**, A representative change in IFNγ-MFI of CD8$^+$ T cells as well as the pooled data of $n = 3$ independent repetitions with 3 technical replicates each is shown. All data in this figure are shown as mean ± s.d. All statistical analysis was done with a two-way ANOVA. **i**,**j**, To evaluate the activation status of anti-HER2 CAR T cells, cells were treated with 500 ng ml$^{-1}$ PGE$_2$ and plated in HER2-precoated wells for 24 h, after which flow cytometry analysis of CD25 (**i**) and CD69 (**j**) expression was performed. Data from $n = 3$ biological donors in 3 technical replicates each are shown as mean ± s.d. Statistical analysis was done using a two-way ANOVA with Tukey's multiple comparison correction.

for little to no HER1 expression[48], we concluded that the system used was dependent on CAR engagement, as anticipated. Overall, this data indicates that EP2$^{-/-}$EP4$^{-/-}$ CAR T cells have a clear killing advantage in PGE$_2$-rich tumours, both in mouse models, as well as patient-derived tumour samples.

## Increased expansion of EP2 and EP4 knockout CAR T cells enhances anti-tumour activity in vivo

To better understand the increased anti-tumour efficacy of EP2$^{-/-}$EP4$^{-/-}$ CAR T cells, we additionally transduced human T cells with a luciferase reporter system (teLuc receptor) to facilitate imaging of adoptively

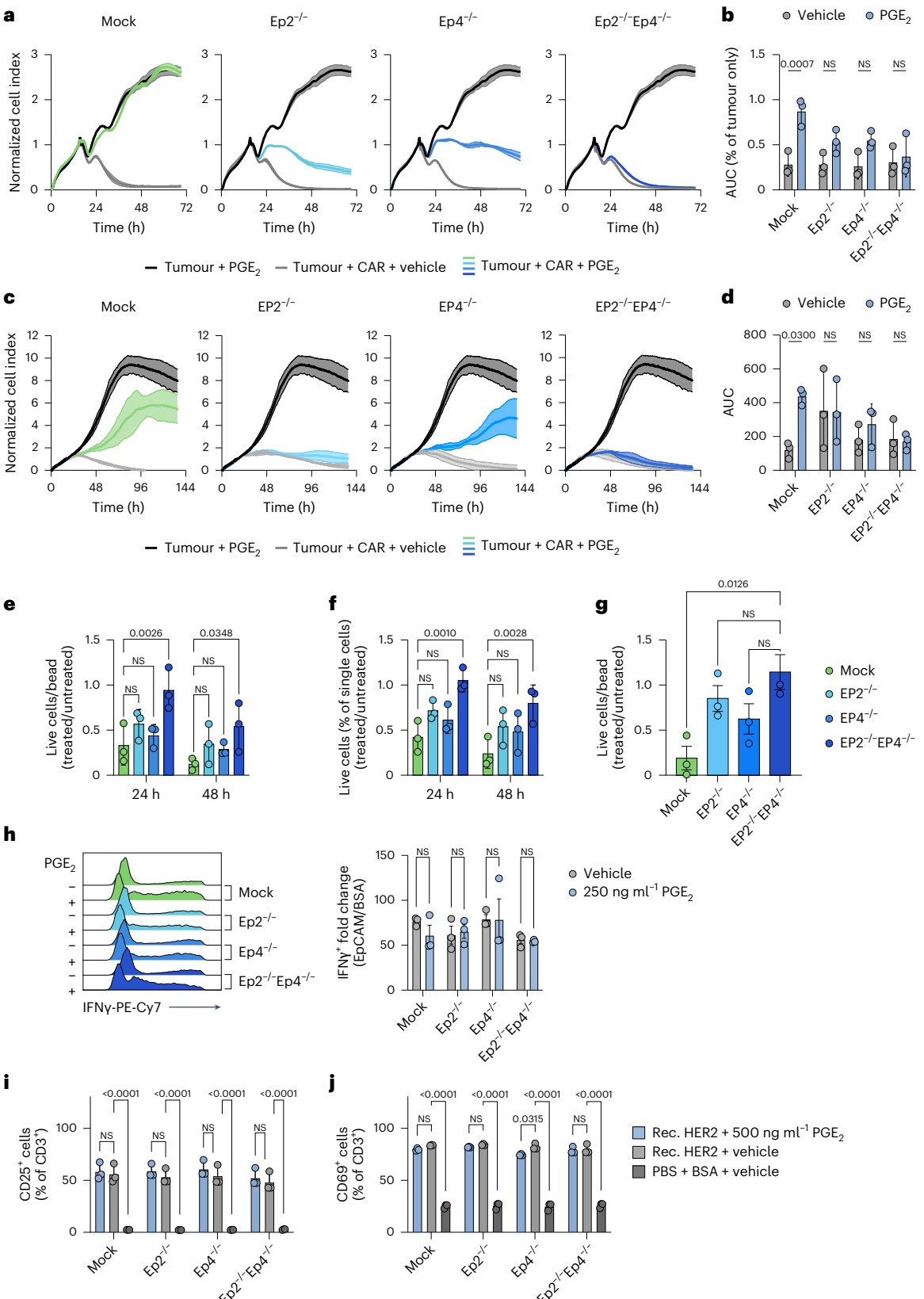

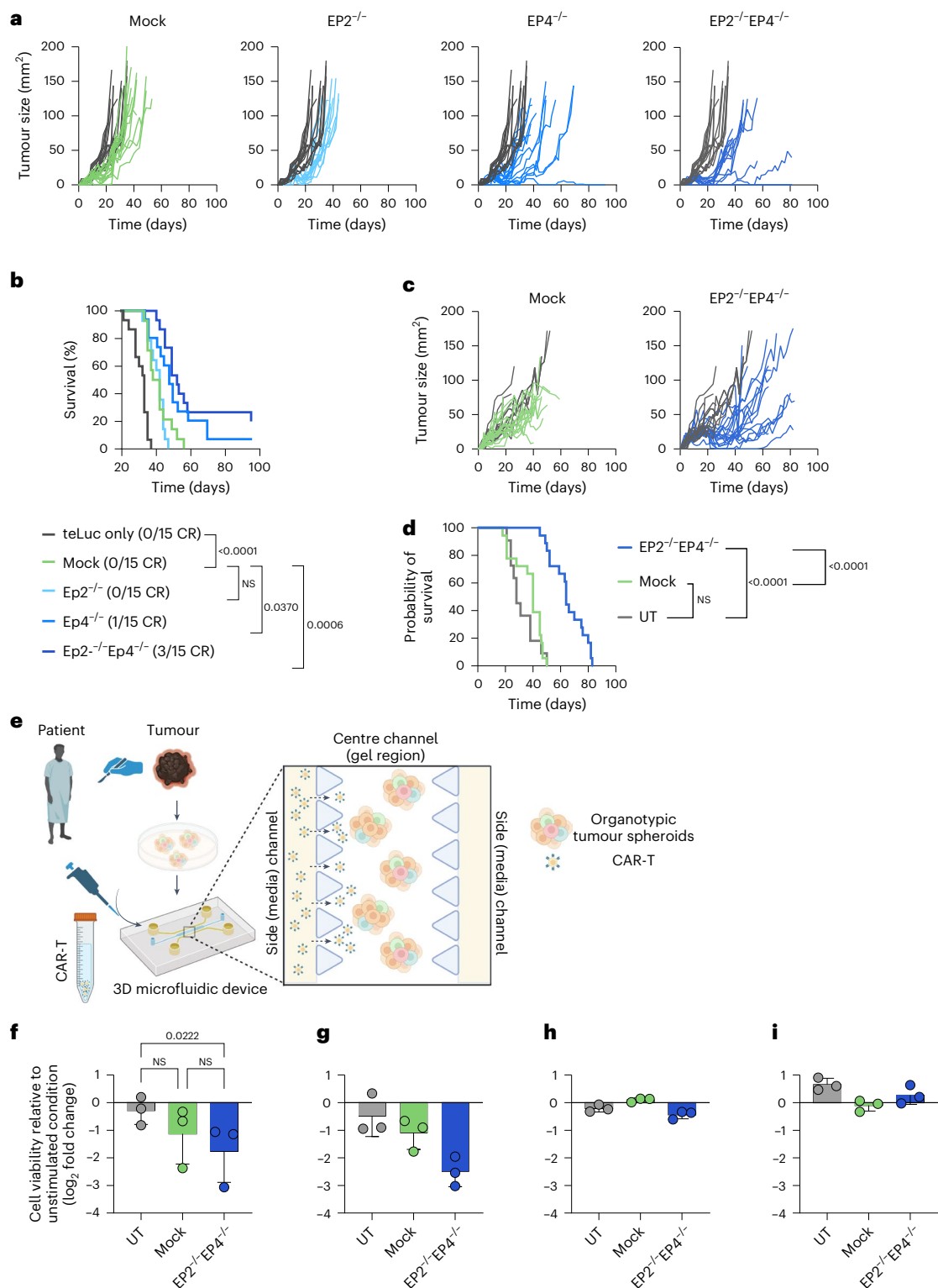

**Fig. 4 | EP2 and EP4 knockout enhances anti-tumour activity of CAR T cells in vivo and in patient-derived samples.** NXG mice were inoculated s.c. with $10^6$ BxPC3 tumour cells and treated with $10^7$ anti-HER2 CAR T cells without EP2$^{-/-}$ and/ or EP4$^{-/-}$. **a,b**, Tumour growth (**a**) and survival (**b**) were monitored for 90 days. Pooled data of $n = 3$ independent repetitions with each $n = 5$ mice per group are shown. Statistical analysis was done with a mixed-effects analysis and log-rank (Mantel−Cox) test. NXG mice were inoculated s.c. with $10^6$ Msto-hMSLN tumour cells and treated with $2 × 10^6$ anti-MSLN CAR T cells without EP2$^{-/-}$ and/or EP4$^{-/-}$. **c,d**, Tumour growth (**c**) and survival (**d**) were monitored for 90 days. Pooled data of $n = 3$ independent repetitions with each $n = 4$–5 mice per group are shown. Statistical analysis was done with a mixed-effects analysis and log-rank

(Mantel−Cox) test. **e**, Schematic representation of workflow. **f**–**i**, PDOTS were prepared in hydrogels composed of rat tail type I collagen and loaded into the centre gel region of the 3D microfluidic culture device, whereafter mock or EP2$^{-/-}$EP4$^{-/-}$ anti-HER1 CAR T cells or UT T cells were added into one of the side channels in the device in an effector to target ratio (E:T) of 3:1. To assess PDOTS viability, dual-label fluorescence live/dead staining was using Hoechst/ propidium iodide staining solution. Viability (mean ± s.d.) of $n = 3$ CRC-derived (**f**), $n = 1$ PDAC-derived (**g**), $n = 1$ neuroendocrine (**h**) and $n = 1$ gastrointestinal cancer tumour-derived (**i**) PDOTS (3 technical replicates each) is shown. Statistical analysis was done with a two-way ANOVA. Illustrations in **e** created with BioRender.com.

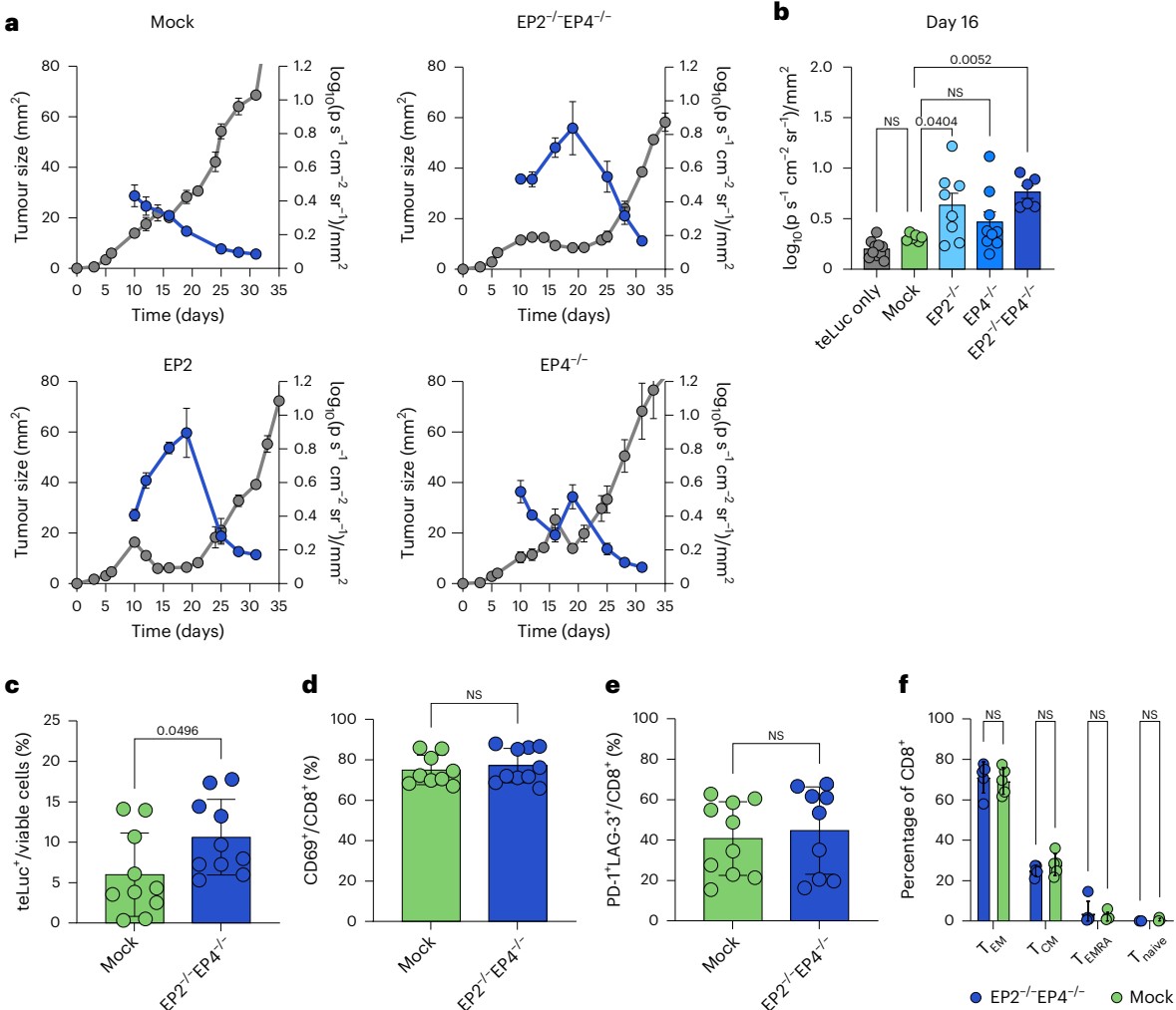

**Fig. 5 | Increased expansion of EP2 and EP4 knockout CAR T cells enhances anti-tumour activity in vivo. a,b**, To track CAR T cell distribution in vivo, NXG mice were inoculated s.c. with $10^6$ BxPC3 tumour cells and treated with $10^7$ anti-HER2 CAR T cells without EP2$^{-/-}$ and/or EP4$^{-/-}$. Anti-HER2 CAR T cells were additionally transduced with a teLuc reporter system, to allow T cell tracking in vivo. Mice were imaged 2× per week by injecting DTZ i.v. before standard IVIS imaging techniques were applied. Log-transformed average radiance was normalized to the tumour area and plotted against the tumour size over time (**a**). Data from a representative experiment with $n = 5$ mice per group are shown. Outliers were identified and excluded. Log-transformed average radiance per tumour area on day 16 post tumour injection pooled from two independent experiments as mean ± s.e.m. (**b**). Outliers were identified and removed; statistics were determined using an ordinary one-way ANOVA. **c–f**, To phenotype tumour-infiltrating CAR T cells, mice were killed 12 days post treatment and tumours were analysed using flow cytometry analysis of teLuc$^+$ cells (**c**), expression of CD69 (**d**) and PD-1 and LAG-3 (**e**). T cell differentiation was determined by the expression of CD62L and CD44 (**f**). Pooled data of $n = 2$ independent experiments with $n = 5$ mice per group are shown. All data in this figure are shown as mean ± s.e.m. Statistical analysis was done with an ordinary one-way ANOVA.

transferred CAR T cells in live animals inoculated with a BxPC3 tumour (Extended Data Fig. 4a,b). Live imaging revealed an expansion of EP2$^{-/-}$ and EP2$^{-/-}$EP4$^{-/-}$ CAR T cells within the first 20-day treatment period. Shortly after, we observed a decline of T cell signal per square millimetre of tumour before outgrowth of the tumour across all treatment groups (Fig. 5a,b and Extended Data Fig. 4b), linking the loss of tumour control to a failure of CAR T cell persistence in the TME. To better understand the reason for tumour relapse, we sought to analyse the tumour of treated mice using flow cytometry analysis, such that we terminated running experiments 12 days post T cell injections. Mice treated with EP2$^{-/-}$EP4$^{-/-}$ anti-HER2 CAR T cells showed increased expansion of therapeutic T cells in tumours (Fig. 5c) compared with groups treated with mock electroporated CAR T cells. Interestingly, no changes in terms of activation, differentiation nor exhaustion phenotype were observed between the two groups (Fig. 5d–f). The improved anti-tumour effect of double-knockout CAR T cells thus stems from an increased expansion of therapeutic cells in the tumour, rather than increased cytotoxicity,

which aligns with our previous observations (Fig. 3h–j). Together with our in vitro characterization, these findings underscore the failure of human CAR T cells to persist in a PGE$_2$-rich TME. Persistence could be rescued by shielding therapeutic cells from PGE$_2$ by knockout of EP2 and EP4, whereby the improvement in persistence was directly linked to an improvement in tumour control and survival.

## Discussion

In this study, we observed the PGE$_2$–EP2–EP4 axis to be a potent inhibitory signalling pathway on CAR T cells in the TME. We therefore propose a novel approach utilizing CRISPR–Cas9-based knockouts of EP2 and EP4 to effectively shield CAR T cells from PGE$_2$ in solid tumours, which may offer the possibility to achieve enhanced CAR T cell persistence and anti-cancer activity within the TME. While conventional EP2/EP4-sufficient CAR T cells exhibited reduced proliferation in PGE$_2$-rich tumours, EP2/EP4-deficient CAR T cells were able to maintain their proliferative capacity and therefore exhibited enhanced anti-tumour

activity. The application of our findings in two different human xenograft in vivo models using NXG mice demonstrated the feasibility of our strategy. Shielding CAR T cells from PGE$_2$ signalling through a simultaneous knockout of EP2 and EP4 increased T cell persistence in the TME and prolonged survival. Interestingly, the activation of CAR T cells remained unaffected by PGE$_2$ not only in vitro but also in in vivo tumour-infiltrating CAR T cells. Furthermore, CRISPR-engineered CAR T cells showed increased anti-tumour activity against patient-derived tumour samples, which underpins the translational potential of the approach.

In a recent study, we uncovered that EP2 and EP4 signalling leads to the abrogation of IL-2 signalling in T cells, caused by downregulation of the IL-2 receptor, thus causing defects in proliferative expansion and persistence of T cells in the TME[40,41]. Interestingly, impaired T cell proliferation was recently confirmed to be caused by PGE$_2$ in the CAR T cell context in an independent study[42]. Furthermore, we could not detect any influence of PGE$_2$ on the migration of CD8$^+$ T cells from the lymph nodes into the tumour in our recent study[40]. This further supports the hypothesis that proliferation and persistence of T cells, rather than an impairment of migration, were causing the observed differences in intratumoural CAR T cell numbers. While other studies reported a reduction of the activation and killing capacity of T cells by PGE$_2$ (refs. [42,49,50]), we did not observe such effects. Rather, our data argue that reduced cytotoxicity is a consequence of impaired T cell expansion. This is important given that most assays used to measure CAR T cell performance focus on analysis of activation or killing capacity through assays such as tumour cell lysis or IFNγ production via enzyme-linked immunosorbent assay (ELISA), without accounting for alterations in effector cell numbers[42,50]. To carefully delineate our observations, we therefore chose methods analysing activation differences at single-cell resolution. In support of this, our recently published sequencing data also did not reveal changes in the T cell activation profiles between T cells deficient and proficient for PGE$_2$ signalling[40]. Both in vitro and in vivo, we did not observe a consistently apparent benefit of the EP2 and EP4 single knockout over the double knockouts across repetitions and donors. This highlights the necessity of the double knockout to completely abrogate PGE$_2$ signalling and guarantee reliable protection from PGE$_2$ signalling in the TME.

COX-dependent PGE$_2$ production is a hallmark of many human cancers[20,37], and limiting the detrimental effects of this inhibitory axis on anti-cancer immunity is a promising strategy to advance T cell-based cancer immunotherapies[40–42]. Therefore, our findings that ablation of EP2 and EP4 on mouse and human therapeutic T cells is technically feasible and achieves durable T cell persistence, leading to anti-cancer activity in tumours, have important clinical implications. After discovering that COX inhibitors had a preventive effect against several cancer types[20,28–30], COX inhibition was tested in a therapeutic setting in combination with chemo- or radiotherapy[25,31–34]. However, side effects such as skin rashes and gastrointestinal bleeding events have led to the discontinuation of the treatment with COX inhibitors in many patients[31–33]. Thus, the shortened duration of treatment was deemed a primary factor for the limited benefits attributed to COX inhibitors in influencing treatment outcomes[31–33]. Such adverse events would also need to be anticipated and eventually overcome in combination with CAR T cells. In addition, a recent in vitro study could show reduced expansion and increased apoptosis of human anti-CD19 CAR T cells upon treatment with physiological concentrations of celecoxib[51]. Together, these observations highlight the need for alternative approaches to shield CAR T cells against PGE$_2$, which highlights the relevance of our approach of ablating EP2 and EP4.

To avoid the drawbacks associated with systemic disruption of homeostatic PGE$_2$ signalling by blocking PGE$_2$ synthesis, selective inhibitors of EP2 and EP4 were developed[52]. The promising preclinical application of these inhibitors in safeguarding T cells from PGE$_2$ prompts a deeper investigation into the strategy of inhibiting PGE$_2$ signalling in therapeutic T cells[35,42,53]. However, EP2 and EP4 are still expressed on a variety of cell types other than T cells, such that concerns related to systemic PGE$_2$ inhibition might still apply. This is limiting the clinical application and leading to the need to further improve this approach. Using our method of knocking out EP2 and EP4, we can selectively protect CAR T cells from PGE$_2$ while leaving systemic PGE$_2$ signalling unaltered. Thereby, we can effectively avoid potential concerns of an antagonist-based approach, such as side effects or dosing issues in peripheral tissues and compartments. Importantly, our approach of EP2 and EP4 knockout has been shown to be effective at physiologically relevant concentrations in vitro and in vivo, as well as in patient-derived samples. Previous studies using small-molecule inhibitors of EP2 and EP4 could only demonstrate benefits at unphysiologically high concentrations of PGE$_2$ and only in vitro[42,50], suggesting that our approach may be more suitable at physiological concentrations of PGE$_2$. Capitalizing on these results, we propose for the first time simultaneously targeting of two G protein-coupled surface receptors (GPCRs EP2 and EP4) that cannot be directly detected by antibodies, rendering efficiency assessment, especially in a space where protein half-life may survive transcription and translation of the corresponding genes, impossible. We postulate for the first time that a functional readout can be used to assess editing efficiencies of therapeutic T cells, opening a new target space for therapeutic engineering.

However, further genetic modification of T cells beyond addition of the CAR harbours its own safety concerns. Enhancing the cytotoxic potential of CAR T cells by releasing checkpoint-like breaks when ablating EP2 and EP4 may raise concerns regarding an unbridled immune response in patients, thereby increasing the potential risk of unwanted side effects, which will need to be carefully weighed against potential benefits[54]. Clinical trials utilizing CRISPR-engineered CAR T cells, such as the knockout of the potent checkpoint inhibitor PD-1 on (CAR) T cells, for example, have not reported increased toxicity compared with trials without such engineering[55]. In addition, multiple engineering steps of T cells before clinical application may raise manufacturing-related safety concerns that require thorough investigation[56,57], which will be crucial for the continued development and application of CRISPR-engineered CAR T cells. In the context of CRISPR-engineered therapeutic cells, the occurrence of unwanted off-target editing possibly causing unforeseeable problems, including oncogenic events, is an additional concern[58–60]. In any case, our whole genome sequencing analysis detected no relevant off-target gene editing effects, indicating that the chosen gRNAs for our protocol exhibit a relatively low-risk profile. Nonetheless, further evaluation, particularly in multiple donors, will be necessary to validate our findings. Notably, no clinical issues regarding the use of CRISPR-engineered cells in patients have been reported so far[55]. In any case, preventive strategies, such as the inclusion of suicide switches, could be implemented if necessary[59,61].

In summary, our work showcases a versatile platform, allowing for the interchange of CAR targets in PGE$_2$-rich tumours. Our findings underscore the relevance of CRISPR-mediated knockouts of EP2 and EP4 in rescuing CAR T cells from PGE$_2$-mediated inhibition, highlighting the potential clinical relevance of CRISPR-mediated receptor knockouts targeting PGE$_2$ for therapeutic applications.

## Methods

### Mice

All mice used in syngeneic experiments carried a C57BL/6J background. OT-I mice (JAX:003831) were used as organ donors for syngeneic therapeutic experiments. *Ptger2$^{-/-}$ Ptger4$^{fl/fl}$* mice were generated by crossing *Ptger2$^{-/-}$* mice (JAX:004376) to *Ptger4$^{fl/fl}$* mice (JAX:028102) and further crossed to *CD4$^{Cre}$* mice (JAX:022071) to generate *CD4$^{Cre}$Ptger2$^{-/-}$Ptger4$^{fl/fl}$* mice. *CD4$^{Cre}$Ptger2$^{-/-}$Ptger4$^{fl/fl}$* mice were further crossed to CD45.1 (JAX:002014) mice and used on a CD45.1/CD45.2 background. For xenograft experiments, NXG mice (NOD-Prkdc$^{scid}$-IL2rg$^{Tm1}$/Rj) were obtained from Janvier. CD45.1/CD45.2 × *CD4$^{Cre}$Ptger2$^{-/-}$Ptger4$^{fl/fl}$* mice

were bred and maintained at the Klinikum Rechts der Isar, Technical University of Munich, TUM. All other mice were maintained and bred at the Klinikum der Universität München. All mice were kept under specific-pathogen-free conditions and in accordance with the guidelines of the Federation of European Laboratory Animal Science Associations. In all experiments, mice at 6–10 weeks of age were randomly assigned to control or treatment groups. Investigators were blinded during data acquisition. Mice were killed by cervical dislocation. Animal experiments were approved by local authorities (Regierung von Oberbayern) and performed in accordance with national guidelines. The experimental protocol mandated the euthanasia of mice upon reaching a critical tumour dimension of 12 mm (syngeneic models) or 15 mm (xenograft models) or ulceration of skin in accordance with established scientific procedures. These criteria were defined as humane surrogate endpoints for animal survival.

## Cell lines

D4M.3A-SIINFEKL cells have been previously described and were cultured in DMEM-F12 medium (Thermo Fisher Scientific)[62], while Panc02 cells were cultured in DMEM medium (Thermo Fisher Scientific), both media supplemented with 10% fetal calf serum (FCS), 1% penicillin, 1% streptomycin and 1% L-glutamine. Panc02 cells overexpressing OVA and EpCAM have been previously described[16,63,64]. BxPC3 and Msto-hMSLN cells were cultured in complete RPMI 1640 containing 10% FCS, 1% penicillin, 1% streptomycin and 1% L-glutamine. The producer cell lines 293VecEco-anti-EpCAM-CAR-mCherry[16], RD114-anti-HER2 CAR, RD114-anti-HER1 CAR and RD114-anti-MSLN CAR[65] were described previously. 293VecEco-OTI-TCRαβ and RD114-teLuc-mCherry producers were generated by expressing the retroviral vector pMP71 carrying the OTI-TCRαβ or teLuc-mCherry sequence in 293Vec-Galv and 293Vec-Eco or RD114 cell lines, as described previously[16,63,64]. In brief, 293Vec-Galv cells were transfected with the respective vector using Lipofectamine 2000. Viral supernatants were taken after 48 h and were used to transduce 293Vec-Eco or 293Vec-RD114 cells in the presence of polybrene. Cells were sorted for transgene-positive cells. All producer cell lines were cultured in complete DMEM medium containing 10% FCS, 1% penicillin, 1% streptomycin and 1% L-glutamine.

## Tumour cell injection and measurement of tumour size

Tumour cell lines were detached by trypsinization (Thermo Fisher Scientific) and washed three times in sterile PBS (Thermo Fisher Scientific). If not stated otherwise, $10^6$ cells were injected subcutaneously (s.c.) in 100 µl sterile PBS into the flank of recipient mice. Tumour growth was measured using a digital caliper, and the tumour area was calculated by multiplying tumour width and length.

## Genetic engineering and culture of primary CAR T cells

Isolation and culture of primary murine and human T cells was done as described previously[15,16,63–65]. In brief, human peripheral blood mononuclear cells (PBMCs) were isolated from healthy donor blood or Buffy-Coats by density gradient centrifugation. CD3⁺ T cells were enriched by MACS using CD3 MicroBeads, human (Miltenyi). Murine T cells were isolated from splenocytes by passing through a cell strainer and removing erythrocytes by using erythrocyte lysis buffer and activated by Dynabeads Mouse T-Activator CD3/CD28. Primary murine CAR T cells were cultured in RPMI 1640 containing 10% FCS, 2 mM L-glutamine and 100 U ml⁻¹ penicillin, 100 mg ml⁻¹ streptomycin, 1% sodium pyruvate, 1 mM HEPES, 50 µM β-mercaptoethanol and 1 mg ml⁻¹ IL-15 in the presence of Dynabeads Mouse T-Activator CD3/CD28. Primary human CAR T cells were cultured in RPMI 1640 containing 2.5% human serum, 1% 2 mM L-glutamine, 100 U ml⁻¹ penicillin, 100 mg ml⁻¹ streptomycin, 1% NEAA, 1% sodium pyruvate, 50 µM β-mercaptoethanol, 1 µg ml⁻¹ IL-2 and 100 µg ml⁻¹ IL-15 in the presence of Dynabeads Human T-Activator CD3/CD28. Retroviral transduction of primary murine and human T cells was done as described previously[15,16,63–65]. In brief, virus was coated onto RetroNectin (Takara)-treated plates and T cells were cultured in virus-coated plates for 24 h. Second-generation CAR T cells with a CD3ζ signalling and a CD28 co-stimulatory domain were used. To obtain EP2 and/or EP4 knockout CAR T cells, CRISPR–Cas9-mediated knockouts were performed 24 h after transduction[15]. Two different Alt-R CRISPR–Cas9 crRNAs (IDT) were used in combination for each of the receptors EP2 (murine, GUAGAAGUAAGGGUACCCGAGUUUUA-GAGCUAUGCU and CCUGCCGCUGCUCAACUACGGUUUUAGAGCU-AUGCU; human, GCGUACGAAGCCAGUACCACGUUUUAGAGCUAUGCU and AGUACGUCCAGUACUGCCCCGUUUUAGAGCUAUGCU) and EP4 (murine, ACAGGCCACCGAAGCUACCGGUUUUAGAGCUAUGCU and CCAGCCGCUUGUCCACGUAGGUUUUAGAGCUAUGCU; human, GGA-GACGACCUUCUACACGCGUUUUAGAGCUAUGCU and CGACUGGACCAC-CAACGUGAGUUUUAGAGCUAUGCU). Alt-R CRISPR tracrRNA (100 µM, IDT) and Alt-R CRISPR–Cas9 crRNA (100 µM, IDT) were hybridized, and RNPs were formed with Alt-R Cas9 Electroporation Enhancer and Alt-R S.p. Cas9 Nuclease V3. Cells were resuspended in 100 µl in electroporation buffer P3 (Lonza) and mixed with the RNPs in a Nucleocuvette Vessel (Lonza). Nucleofection was performed in a 4D Nucleofector (Lonza) with the pulse programme CM137 (murine T cells) or EH115 (human T cells). To identify editing efficiencies for the genes of interest, polyclonal T cell knockout pools were lysed, barcoded using dual PCR barcoding and subsequently subjected to deep sequencing using the MiSeq (Illumina) sequencing system, as described before[66]. In brief, cells were lysed and subjected to dual PCR barcoding (primers: EP2 gRNA1: Fwd, CTGGGGAACCTCATAGCACT; Rev, GAAGAAGGTCATGGCGAAAG

> EP2 gRNA2: Fwd, GCACCCCTACTTCTACCAGC; Rev, GCATGCGG ATGAGGTTGAGA
> EP2 gRNA1 + gRNA2: Fwd, CTCTCCTTGTTCCACGTGCT; Rev, CATGG ACACCCTTTCCCCTC
> EP4 gRNA1: Fwd, CTGAACAGCCCAGTGACCAT; Rev, CCGGAC AGGCTGAAGAAGAG
> EP4 gRNA2: Fwd, TATGCGTCCAACGTGCTCTT; Rev, ATGAACTG GCGGTGCATGC
> EP4 gRNA1 + gRNA2: AATTCGTCCGCCTCCTTGAG; Rev, ATGAAC TGGCGGTGCATGC; adapter sequences: Fwd-Primer, ACACTCT TTCCCTACACGACGctcttccgatct

Rev-Primer, TGACTGGAGTTCAGACGTGTGctcttccgatct), followed by gel purification and Illumina MiSeq deep sequencing. Sequencing data were processed using OutKnocker 2.0 software for analysis.

## Analysis of CRISPR off-target editing by whole genome sequencing

EP2⁻/⁻, EP4⁻/⁻, EP2⁻/⁻EP4⁻/⁻ and mock anti-HER2-CAR T cells, as well as UT control T cells, were generated as described above. Genomic DNA was isolated from polyclonal T cell pools using the DNeasy blood and tissue kit (Quiagen), and whole genome sequencing with 30× coverage was performed by Novogene, including mapping to the reference genome hg38 and variant calling based on GATK. Off-targets were predicted in silico using Cas-OFFinder allowing up to four mismatches[43]. All potential off-target sites were manually inspected in our whole genome sequencing data in the IGV. In addition, similar as described by others[67], GATK-called variants absent from the control samples (untransduced T cells and mock anti-HER2-CAR T cells), but reproducible across replicates, were used for unbiased off-target identification. All variants classified as 'high priority' (in brief, variants located in an exon or splicing region that naturally occur in less than 1% of genomes of the 1000 Genomes Project and predicted changes on protein level) were manually evaluated in the IGV.

## Adoptive T cell transfer

For T cell transfers in the syngeneic tracking experiment, $5 \times 10^6$ CD45.1 × *CD4^Cre Ptger2^−/− Ptger4^fl/fl* and $5 \times 10^6$ CD90.1 T cells transduced with OT-I were co-injected intravenously (i.v.) in 100 µl sterile PBS into wild-type

(WT) recipient mice with s.c. tumours as indicated in the figure legends. For syngeneic therapeutic experiments, OT-I T cells with mock, EP2 and/or EP4 knockout (done by CRISPR–Cas9) were injected i.v. in 100 µl sterile PBS into WT recipient mice with s.c. tumours 6–7 days post s.c. tumour injection. For T cell transfer into NXG mice, $10^7$ human anti-HER2 CAR T or $2 \times 10^6$ human anti-MLSN CAR T cells with mock or EP2 and EP4 knockout (done by CRISPR–Cas9) were injected i.v. in 100 µl sterile PBS into female NXG recipient mice 5–8 days post s.c. tumour injection.

### In vivo T cell tracking
To allow T cell tracking in vivo, human anti-HER2 CAR T cells were co-transduced with teLuc. When measuring bioluminescence imaging (BLI), mice were injected i.v. with a stock solution of DTZ (1 mg ml$^{-1}$ diphenylterazine (DTZ) (MedChem) solved in 10% DMF (Carl Roth), 40% PEG-300 (Carl Roth), 20% Tween-20 (Carl Roth) and 30% NaCl (B. Braun SE)) diluted 1:3 in PBS directly before use. DTZ (0.07 mg) was injected per mouse, and mice were anaesthetized using an isoflurane–oxygen mixture (1.5–2.5%)[68]. An in vivo imaging system platform Lumina X5 (IVIS, PerkinElmer) was used to measure the BLI signal.

### Cell isolation for flow cytometry
Tumours, lymph nodes or spleens of tumour-bearing mice were excised at the indicated time points after T cell transplantation. Tumour samples were mechanically dissociated and incubated with Collagenase IV (1 mg ml$^{-1}$, Sigma-Aldrich) and DNase I (0.05 mg ml$^{-1}$, Sigma-Aldrich) for 30 min at 37 °C and filtered through a 70 µm and a 30 µm cell strainer (Miltenyi) to generate single-cell suspensions. Spleens were passed through a 70 µm cell strainer, followed by red blood cell lysis and a second filtration step using a 30 µm cell strainer. Lymph nodes were passed through a 30 µm cell strainer. Cells were subsequently submitted to further extra- and intracellular staining for flow cytometry.

### Flow cytometry
Extracellular staining was performed for 15 min at 4 °C in PBS. The following antibodies and staining reagents were used for flow cytometry or cell sorting: Fixable Viability Dye eFluor 780 (1:1,000, Invitrogen), APC anti-human CD3 (1:100, clone OKT3, BioLegend), Pacific Blue anti-mouse CD3 (1:100, clone 17A2, BioLegend), Pacific Blue anti-mouse CD4 (1:100, clone GK1.5, BioLegend), FITC anti-mouse CD4 (1:100, clone GK1.5, BioLegend), PE-Cy7 anti-human CD4 (1:100, clone OKT4, BioLegend), PerCP anti-human CD8 (1:100, clone HIT8a, BioLegend), Pacific Blue anti-mouse CD8a (1:100, clone 53-6.7, BioLegend), FITC anti-mouse CD8 (1:100, clone 53-6.7, BioLegend), BV605 anti-human CD8 (1:100, clone SK1, BioLegend), BV785 anti-mouse CD8 (1:100, clone 53-6.7, BioLegend), Alexa Fluor 647 mouse anti-CREB (pS133)/ATF-1 (pS63) (1:20, clone J151-21, BD Biosciences), BV650 anti-human CD69 (1:100, clone FN50, BioLegend), PE/Cy7 anti-human CD62L (1:100, clone DREG-56, BioLegend), BV510 anti-human CD44 (1:100, clone IM7, BioLegend), Alexa Fluor 700 anti-human PD-1 (1:100, clone EH12.2H7, BioLegend), BV510 anti-human TIM-3 (1:100, clone F38-2E2, BioLegend), APC anti-human LAG-3 (1:100, clone 7G2C65, BioLegend), PE-Cy7 anti-mouse IFNγ (1:100, clone XMG1.2, eBioscience), BV421 anti-human CD45 (1:100, clone 2D1, BioLegend), BV605 anti-human CD4 (1:100, clone UCHL1, BioLegend), Alexa Fluor 700 anti-mouse CD4 (1:100, clone GK1.5, BioLegend), BV711 anti-mouse CD45.1 (1:100, clone A20, BioLegend), APC anti-rat CD90/mouse CD90.1 (Thy-1.1) (1:100, clone OX-7, BioLegend), BV421 anti-human CD69 (1:100, clone F50, BioLegend), PerCP-Cy5.5 anti-human CD25 (1:100, clone BC96, BioLegend) and FITC anti-human/mouse/rat c-myc (1:100, clone SH1-26E7.1.3, Miltenyi).

### Patient specimens
Patient tumour samples were collected and analysed according to Dana-Farber/Harvard Cancer Center (DF/HCC) Institutional Review Board (IRB)-approved protocols. Written informed consent was obtained from all subjects, and a cohort of patients treated at

Massachusetts General Hospital was assembled for PDOTS profiling. These studies were conducted according to the Declaration of Helsinki and approved by the DF/HCC IRB.

### PDOTS preparation, culture and viability assessment
PDOTS were prepared and cultured, as previously described[69]. In brief, PDOTS were prepared in hydrogels composed of rat tail type I collagen (Corning, catalogue number 35426) at a final concentration of 1.7 mg ml$^{-1}$. Spheroid–collagen mixtures (10 µl) were counted with a Countess II (Invitrogen) and loaded into the centre gel region of the 3D microfluidic culture device (Dax-01, AIM Biotech), and after incubation (30 min at 37 °C, 5% CO$_2$) in sterile humidity chambers. Anti-HER1 CAR T cells were added into one of the side channels in the device, and then collagen hydrogels containing tumour spheroids/PDOTS were hydrated with media (RPMI supplemented with 10% FBS and 1% penicillin–streptomycin), in the presence or absence of the indicated treatments. To assess PDOTS viability, dual-label fluorescence live/dead staining was using Hoechst/propidium iodide staining solution (Nexcelom, CSK-V0005) (19,20). Images were obtained following incubation with the stain solution (30 min, 37 °C, 5% CO$_2$), and image capture and analysis were performed using a Nikon Eclipse NiE fluorescence microscope equipped with a motorized stage (ProScan) and ZYLA4.2 Plus USB3 Camera (Andor) and NIS-Elements AR software package (Nikon). Live and dead cells were quantified by measuring the total raw cell area for each dye. Percent change and log$_2$FC (L2FC) data were generated using raw fluorescence data (live) for given treatments relative to control conditions.

### Analysing CREB phosphorylation
For the pCREB staining (BD Phosphoflow AF647 anti-pCREB, BD Biosciences), murine anti-EpCAM CAR or human anti-HER2 CAR T cells were surface stained and stimulated with 1,600 ng ml$^{-1}$ (murine) and 2,000 ng ml$^{-1}$ (human) for 60 min at 37 °C. Phosphorylation was terminated by fixing the cells with BD Cytofix (BD Biosciences) for 10 min at 37 °C and subsequently permeabilization using BD Phosflow Perm Buffer III (BD Biosciences) for 30 min on ice, before intracellular staining[70].

### CAR T cell expansion analysis
For the EdU incorporation assay, murine anti-EpCAM CAR T cells were stimulated with 250 ng ml$^{-1}$ PGE$_2$. EdU incorporation was measured after 24 h and 48 h. To determine the in vitro expansion murine anti-EpCAM CAR T cells or human anti-HER2 CAR T cells, cells were treated with 250 ng ml$^{-1}$ PGE$_2$ (murine) or 500 ng ml$^{-1}$ PGE$_2$ (human), stained with Fixable Viability Dye eFluor 780 (1:1,000, Invitrogen) and counted by using Countbright absolute counting beads after 72 h.

### CAR T cell activation analysis
For the activation assay, 96-well plates were coated with 1 µg ml$^{-1}$ recombinant Fc-tagged EpCAM (R&D Systems) or human recombinant HER2/ERBB2 protein (Sino Biological) overnight and blocked with 1% BSA. Murine anti-EpCAM CAR T cells or human anti-HER2 CAR T cells were activated on the respective antigen-coated plates in the presence of 250 ng ml$^{-1}$ PGE$_2$ (murine) or 500 ng ml$^{-1}$ PGE$_2$ (human) for 24 h. After that, cells were submitted to cell surface staining for CD25 and CD69 or intracellular staining for IFNγ (refer to flow cytometry).

### Measuring of cAMP levels
cAMP production was measured by stimulating murine anti-EpCAM CAR or human anti-HER2 CAR T cells with 2,000 ng ml$^{-1}$ PGE$_2$ for 60 min at 37 °C and subsequently quantifying cAMP levels with the cAMP-Glo Assay Kit (Promega) according to the manufacturer's instructions.

### Determining the killing capacity of CAR T cells
Killing capacity was determined by co-culturing murine anti-EpCAM CAR T cells with Panc02-OVA-EpCAM cells (1:1 effector-to-target ratio)

or human anti-HER2 CAR T cells with BxPC3 (1:4–5 effector-to-target ratio) pre-incubated with PGE$_2$ for 48 h (250 ng ml$^{-1}$ for murine and 500 ng ml$^{-1}$ for human CAR T cells) while measuring the impedance using the xCELLigence-RTCA system.

## Statistical analyses

The Prism software (v10.0.3 GraphPad) was used for all statistical analyses. Statistical significance for the comparisons among experimental groups was assessed using a one- or two-way analysis of variance (ANOVA) as indicated in the figure legends. Tumour growth profiles were assessed with mixed-effects analyses, and for survival analyses, a log-rank (Mantel–Cox) test was used. Data are shown as mean ± s.d. or s.e.m., as indicated in the figure legends. Significance was assumed with *$P$ < 0.05, **$P$ < 0.01, ***$P$ < 0.001 and ****$P$ < 0.0001. Imaging data were log-transformed to achieve a normal distribution before statistical analysis. Outliers were identified using the robust regression and outlier removal (ROUT) method with ROUT-coefficient $Q$ = 2%.

## Reporting summary

Further information on research design is available in the Nature Portfolio Reporting Summary linked to this article.

## Data availability

The main data supporting the results in this study are available within the paper and its extended data. The raw datasets generated in this study are available via the Open Data LMU repository at https://data.ub.uni-muenchen.de/713/ in accordance with the principles of open science of the European Research Council (ERC). Whole genome sequencing data will be made available for research purposes from the corresponding author on reasonable request and regulatory clearance.

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

## Acknowledgements

This study was supported by the Förderprogramm für Forschung und Lehre der Medizinischen Fakultät der LMU (grant number 1138 to A.G.), the Bavarian Cancer Research Center (BZKF) (A.G. and TANGO to S.K.), the Deutsche Forschungsgemeinschaft (DFG, grant number GO 3823/1-1 to A.G.; grant numbers KO5055-2-1 and KO5055/3-1 to S.K.; grant numbers 449174900 and 461704785 (SPP) to J.P.B.), the International Doctoral Program 'i-Target: immunotargeting of cancer' (funded by the Elite Network of Bavaria; to S.K. and S.E.), the Melanoma Research Alliance (grant number 409510 to S.K.), Marie Sklodowska-Curie Training Network for Optimizing Adoptive T Cell Therapy of Cancer (funded by the Horizon 2020 programme of the European Union; grant 955575 to S.K.), Marie Sklodowska-Curie Training Network for tracking and controlling therapeutic immune cells in cancer (funded by the Horizon Programme of the EU, grant 101168810 to S.K.), Else Kröner-Fresenius-Stiftung (IOLIN to A.G., A.O., I.P., S.M., S. Stock and S.K.), German Cancer Aid (AvantCAR.de and grant number 70117182 to S.K.), the Wilhelm-Sander-Stiftung (to S.K.), Ernst Jung Stiftung (to S.K.), Institutional Strategy LMUexcellent of LMU Munich (within the

framework of the German Excellence Initiative; to S.E. and S.K.), the Go-Bio-Initiative (to S.K.), the m4-Award of the Bavarian Ministry for Economical Affairs (to S.E. and S.K.), Bundesministerium för Bildung und Forschung (to S.E. and S.K.), the EUROSTAR-Programm (to A.O.), European Research Council (starting grant 756017, PoC grant 101100460 and CoG 101124203 to S.K.), by the SFB-TRR 338/ 1 2021–452881907 (to S.K.), Fritz-Bender Foundation (to S.K.), Deutsche José Carreras Leukämie Stiftung (to S.K.), Hector Foundation (to S.K.), Bavarian Research Foundation (BAYCELLATOR to S.K.), the Monika-Kutzner Foundation (to M.P.T. and S.K.), the Bruno and Helene Jöster Foundation (360° CAR to S.K.), the Dr. Rurainski-Foundation (to S.K.) and the Constanze and Dr. Brigitte Wegener Foundation. L.G. received funding from the Studienstiftung des Deutschen Volkes. S. Stock was supported by the Else Kröner-Fresenius-Stiftung, the Munich Clinician Scientist Program (MCSP), the DKTK School of Oncology, the Novartis InCa-Förderpreis 2022 for young researchers, the SFB-TRR 338, the Förderprogramm för Forschung und Lehre (grant number 1168) and the Momente Mentoring Program of the Medical Faculty of the LMU Munich. The Jenkins Lab acknowledges support for the MGH Tumor Cartography Center through the Massachusetts Life Sciences Center Research Infrastructure Program. J.D. received funding supporting this study from the Deutsche Gesellschaft für Immun- und Targeted Therapie. The funders had no role in study design, data collection and analysis, decision to publish or preparation of the manuscript.

## Author contributions

Conceptualization: J.D., L.G., S.B.L., J.P.B. and S.K. Investigation: J.D., L.G., S.B.L., A.O., Y.S., I.P., L.F., S. Spajic, S. L., S.M., L.M., D.B., D.S., D.A.S., E.C., V.H., M.S., A.G., N.S., S. Stock, A.H., P.J.M., T.S., M.P.T., S.C. and G.M.B. Formal analysis: J.D., L.G. and S.K. Supervision: S.E., R.W.J., S.K. and J.P.B. Writing: J.D., L.G., S.B.L., J.P.B. and S.K. All authors contributed to feedback and proofreading. All authors approved the paper for publication.

## Funding

## Competing interests

S.K. has received honoraria from Plectonic, TCR2 Inc., Regeneron, Miltenyi, Galapagos, Cymab, Novartis, BMS and GSK. S.K. is an inventor of several patents in the field of immuno-oncology. S.K. and S.E. received licence fees from TCR2 Inc. and Carina Biotech. S.K. received research support from TCR2 Inc., Tabby Therapeutics, Catalym GmbH, Plectonic GmbH and Arcus Bioscience for work unrelated to the paper. R.W.J. is a member of the advisory board for and has a financial interest in Xsphera Biosciences Inc., a company focused on using ex vivo profiling technology to deliver functional, precision immune-oncology solutions for patients, providers and drug development companies. R.W.J. has received honoraria from Incyte (invited speaker), G1 Therapeutics (advisory board) and Bioxcel Therapeutics (invited speaker). R.W.J. has an ownership interest in US patents US20200399573A9 and US20210363595A1. R.W.J.'s interests were reviewed and are managed by Massachusetts General Hospital and Mass General Brigham in accordance with their conflict-of-interest policies. G.M.B. has sponsored research agreements through her institution with Olink Proteomics, Teiko Bio, InterVenn Biosciences, Palleon Pharmaceuticals, Astellas and AstraZeneca. She served on advisory boards for Iovance, Merck, Moderna, Nektar Therapeutics, Novartis, Replimune and Ankyra Therapeutics. She consults for Merck, InterVenn Biosciences, Iovance and Ankyra Therapeutics. She holds equity in Ankyra Therapeutics. The remaining authors declare no competing interests.

## Additional information

**Extended data** is available for this paper at https://doi.org/10.1038/s41551-025-01610-6.

**Correspondence and requests for materials** should be addressed to Sebastian Kobold.

[1]Institute of Clinical Pharmacology, LMU University Hospital, LMU Munich, Munich, Germany. [2]Institute of Molecular Immunology, TUM University Hospital, School of Medicine and Health, Technical University of Munich (TUM), Munich, Germany. [3]Servier Deutschland, Munich, Germany. [4]Mass General Cancer Center, Krantz Family Center for Cancer Research, Department of Medicine, Massachusetts General Hospital, Harvard Medical School, Boston, MA, USA. [5]Broad Institute of MIT and Harvard, Cambridge, MA, USA. [6]Department of Medicine II, LMU University Hospital, Munich, Germany. [7]Gene Center, Department of Biochemistry, LMU Munich, Munich, Germany. [8]Department of Medicine III, LMU University Hospital, LMU, Munich, Germany. [9]Division of Gastrointestinal and Oncologic Surgery, Department of Surgery, Massachusetts General Hospital, Harvard Medical School, Boston, MA, USA. [10]German Cancer Consortium (DKTK), Partner Site Munich, a partnership between the DKFZ Heidelberg and the LMU University Hospital, LMU, Munich, Germany. [11]Einheit für Klinische Pharmakologie (EKLiP), Helmholtz Zentrum München—German Research Center for Environmental Health Neuherberg, Munich, Germany. [12]M3 Research Center, University Hospital Tübingen, University of Tübingen, Tübingen, Germany. [13]Department of Experimental Immunology, Institute of Immunology, University of Tübingen, Tübingen, Germany. [14]German Center for Lung Research (DZL), Partner Site Munich, Munich, Germany. [15]These authors contributed equally: Janina Dörr, Lisa Gregor, Sebastian B. Lacher. [16]These authors jointly supervised this work: Jan P. Böttcher, Sebastian Kobold. ✉e-mail: Sebastian.kobold@med.uni-muenchen.de

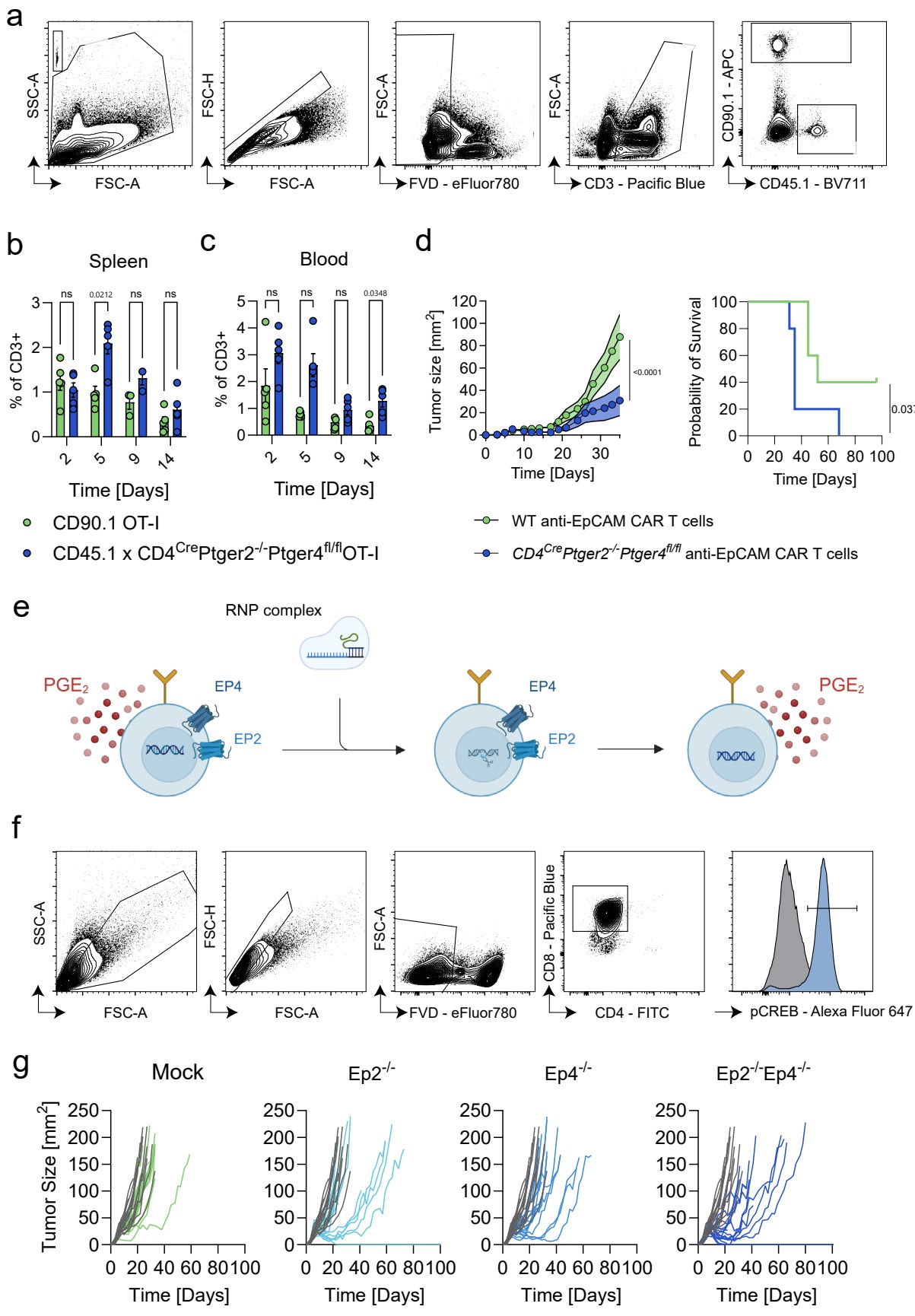

**Extended Data Fig. 1 | See next page for caption.**

**Extended Data Fig. 1 | Ep2 and Ep4 knockout improves the efficacy of adoptively transferred OT-I T cells.** (**a**) 5 ×10⁶ CD45.1 x *CD4^{Cre}Ptger2^{-/-} Ptger4^{fl/fl}* OT-I T cells and 5 ×10⁶ CD90.1 OT-I T cells were co-injected into D4M.3A-SIINFEKL (10⁶ cells s. c.) bearing mice. On days 2, 5, 9 and 14 the abundance of T cell populations was analyzed using flow cytometry; gating strategy depicted. T cell populations in spleen (**b**) and blood (**c**) were analyzed (data shown as mean ± SEM of n = 5 mice per group). Statistical analysis was done with a two-way ANOVA. (**d**) 10 ×10⁶ *CD4^{Cre}Ptger2^{-/-} Ptger4^{fl/fl}* anti-EpCAM-CAR T cells and 10 ×10⁶ anti-EpCAM-CAR T cells were intravenously injected into Panc02-OVA-EpCAM (2 ×10⁶ cells s. c.) bearing mice. Tumor growth and survival are depicted for

n = 5 mice per group. (**e**) Schematic representation of Ep2⁻/⁻Ep4⁻/⁻ OT-I T cells production workflow. OT-I T cells were submitted to genetic editing by inserting the RNP complex through electroporation. (**f**) OT-I T cells with Ep2 and/or Ep4 knockout were treated with 2000 ng/ml PGE₂. CREB phosphorylation was measured by flow cytometry. Depicted is the gating strategy for flow cytometry analysis. (**g**) 10⁷ OT-I splenocytes with Ep2 and/or Ep4 knockout were adoptively transferred (i. v.) into recipient mice bearing D4M.3A-SIINFEKL tumors (10⁶ cells s. c.). Single tumor growth curves of n = 3 independent repetitions with n = 5 mice per group are shown. Illustrations in **e** created with BioRender.com.

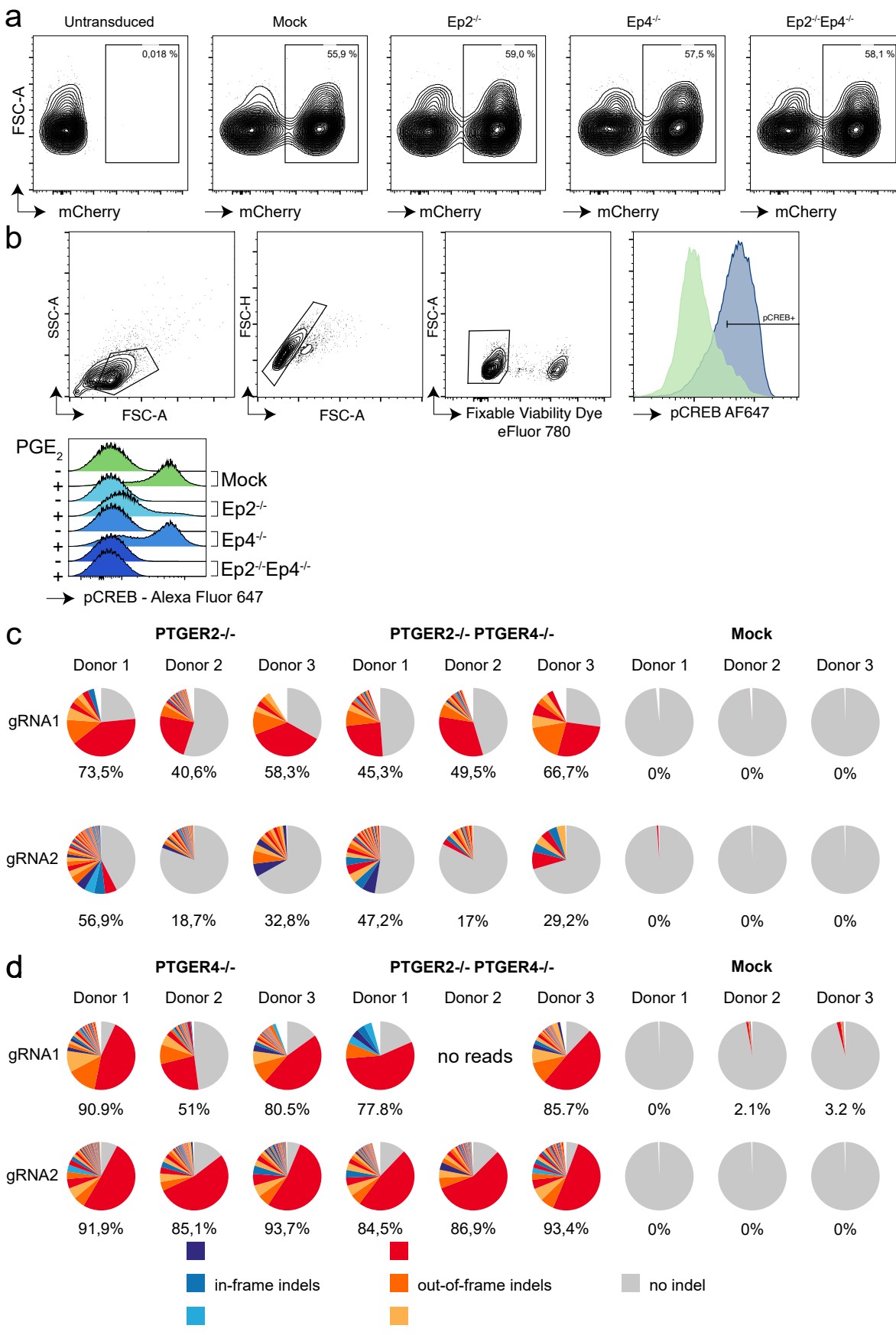

**Extended Data Fig. 2 | See next page for caption.**

**Extended Data Fig. 2 | Generation of CAR T cells with a CRISPR-based knockout of EP2 and EP4. (a**) Representative transduction efficiencies for murine T cells with the different Ep2 and Ep4 knockouts with the anti-EpCAM-CAR measured using flow cytometry by its mCherry-tag. Genomic DNA samples from human anti-HER2 CAR T cells with their respective EP2 and EP4 knockouts were submitted to MiSeq Illumina Sequencing. Percentages of in- and out-of-frame indels are shown for n = 3 human donors at the PTGER2 locus (**b**) and the PTGER4 locus (**c**) for both gRNAs. (**d**) T cells with EP2 and/or EP4 knockout were treated with 1600 ng/ml (murine) or 2000 ng/ml PGE$_2$ (human). CREB phosphorylation was measured by flow cytometry. Depicted is the gating strategy for flow cytometry analysis and a representative result for all knockout conditions.

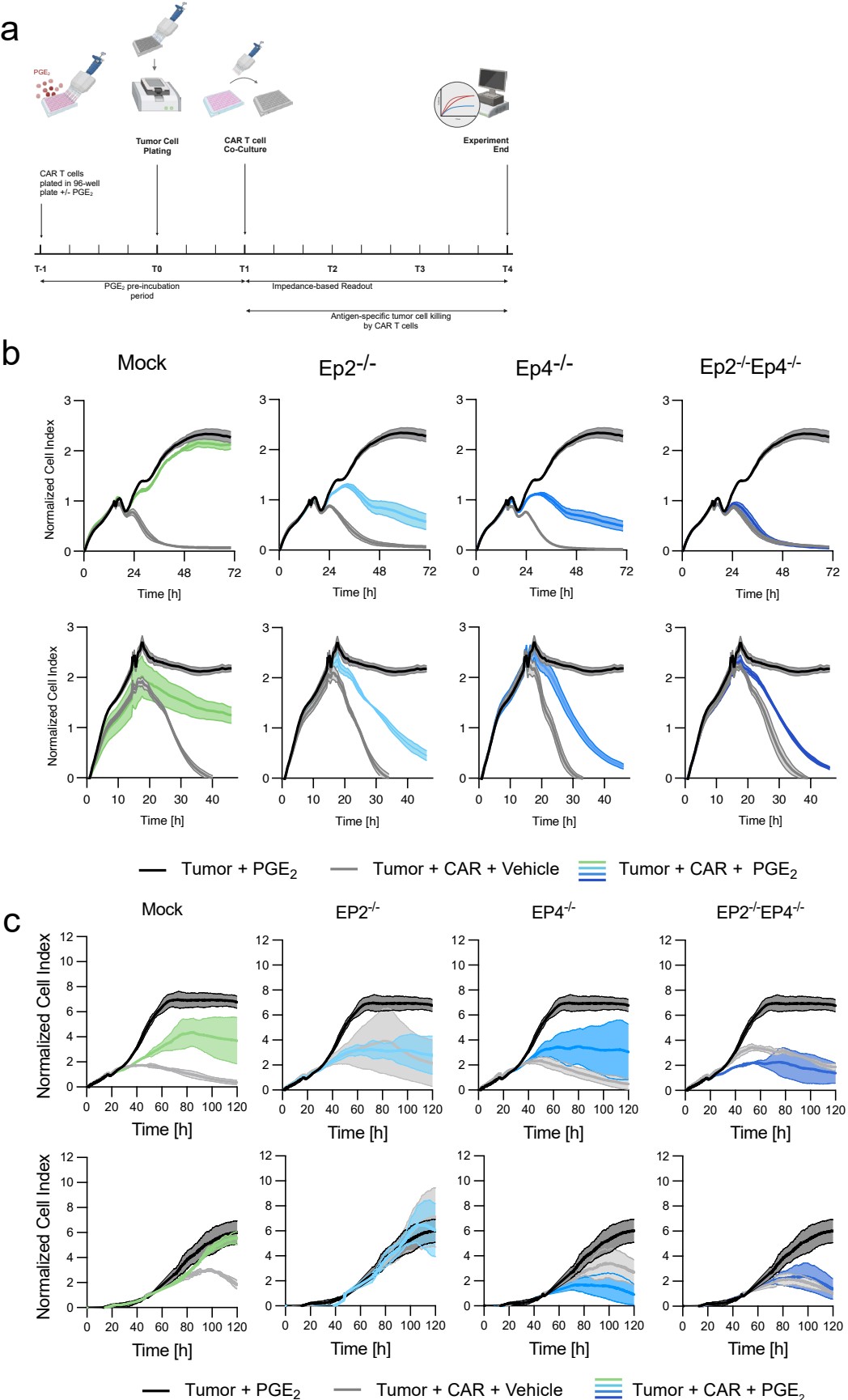

**Extended Data Fig. 3 | See next page for caption.**

**Extended Data Fig. 3 | Improved expansion and survival of EP2-/-EP4-/- CAR T cells in the presence of PGE2 leads to enhanced anti-tumor function in vitro.** (**a**) Schematic representation of the co-culture setup. Data from n = 2 independent experiments with three technical replicates each (mean ± SD), each row of graphs represents one independent donor. CAR T cells were pretreated with 250 ng/ml (murine, (**b**)) and 500 ng/ml (human (**c**)) PGE$_2$ for 48 h, after which co-culture experiments with Panc02-OVA-EpCAM (murine) or BxPC3 (human) cells were performed. Target killing was measured using impedance-based readout. Illustrations in **a** created with BioRender.com.

## a

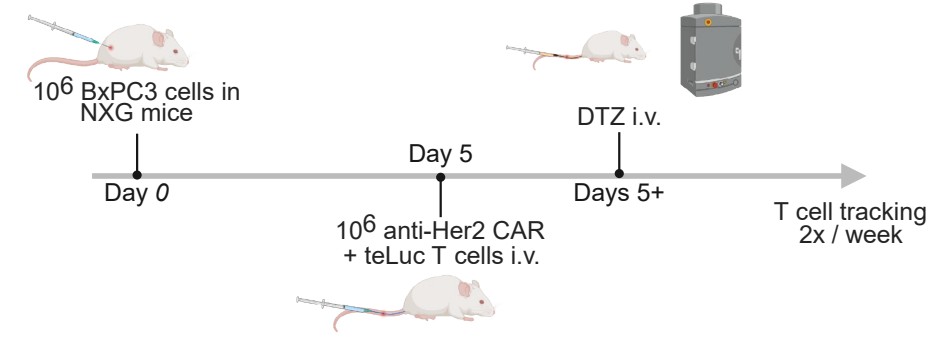

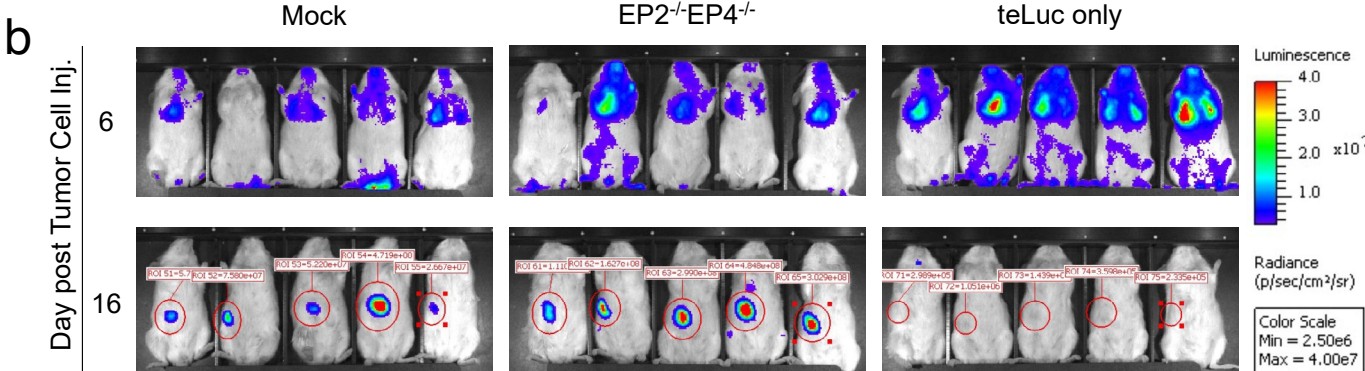

**Extended Data Fig. 4 | Increased expansion of EP2 and EP4 knockout CAR T cells enhances anti-tumor activity in vivo.** (**a**) To track CAR T cell distribution *in vivo*, NXG mice were inoculated s. c. with $10^6$ BxPC3 tumor cells and treated with $10^7$ anti-HER2 CAR T cells w/o EP2$^{-/-}$ and/or EP4$^{-/-}$. Anti-HER2 CAR T cells were additionally transduced with a teLuc reporter system to allow T cell tracking *in vivo*. Mice were imaged 2 x / week by injecting DTZ i. v. before standard IVIS imaging techniques were applied. (**b**) Representative images of luminescence on day 16 post tumor injection, n = 5 mice per group. Panel **a** created with BioRender.com.

# Reporting Summary

## Statistics

For all statistical analyses, confirm that the following items are present in the figure legend, table legend, main text, or Methods section.

| n/a | Confirmed | |
|---|---|---|
| ☐ | ☒ | The exact sample size (*n*) for each experimental group/condition, given as a discrete number and unit of measurement |
| ☐ | ☒ | A statement on whether measurements were taken from distinct samples or whether the same sample was measured repeatedly |
| ☐ | ☒ | The statistical test(s) used AND whether they are one- or two-sided<br>*Only common tests should be described solely by name; describe more complex techniques in the Methods section.* |
| ☒ | ☐ | A description of all covariates tested |
| ☐ | ☒ | A description of any assumptions or corrections, such as tests of normality and adjustment for multiple comparisons |
| ☐ | ☒ | A full description of the statistical parameters including central tendency (e.g. means) or other basic estimates (e.g. regression coefficient) AND variation (e.g. standard deviation) or associated estimates of uncertainty (e.g. confidence intervals) |
| ☐ | ☒ | For null hypothesis testing, the test statistic (e.g. *F*, *t*, *r*) with confidence intervals, effect sizes, degrees of freedom and *P* value noted<br>*Give P values as exact values whenever suitable.* |
| ☒ | ☐ | For Bayesian analysis, information on the choice of priors and Markov chain Monte Carlo settings |
| ☒ | ☐ | For hierarchical and complex designs, identification of the appropriate level for tests and full reporting of outcomes |
| ☒ | ☐ | Estimates of effect sizes (e.g. Cohen's *d*, Pearson's *r*), indicating how they were calculated |

*Our web collection on statistics for biologists contains articles on many of the points above.*

## Software and code

Policy information about availability of computer code

| | |
|---|---|
| Data collection | LRSFortessa Cell Analyzer and FACS Canto II Flow Cytometer - BD FACSDiva (BD Biosciences)<br>CytoFLEX LX, - CytExpert (Beckmann Coulter)<br>xCELLigence RTCA MP&SP - RTCA Software Pro Agilent<br>Berthold Tristar 3 - MikroWin<br>IVIS Lumina X5, Perkin Elmer - Living Image 4.4., Perkin Elmer<br>MiSeq Illunina, Illumina<br>Eclipse NiE fluorescent microscope, Nikon |
| Data analysis | Excel (Microsoft Office Professional Plus 2016), FlowJo (BD Biosciences), Prism (Graphpad), CytExpert v(Beckmann Coulter), RTCA Software Pro (Agilent), AffinityDesigner (Serif), IVIS Lumina X5, Perkin Elmer - Living Image 4.4., Perkin Elmer, NIS-Elements AR software (Nikon) |

For manuscripts utilizing custom algorithms or software that are central to the research but not yet described in published literature, software must be made available to editors and reviewers. We strongly encourage code deposition in a community repository (e.g. GitHub). See the Nature Portfolio guidelines for submitting code & software for further information.

## Data

Policy information about availability of data

All manuscripts must include a data availability statement. This statement should provide the following information, where applicable:

- Accession codes, unique identifiers, or web links for publicly available datasets
- A description of any restrictions on data availability
- For clinical datasets or third party data, please ensure that the statement adheres to our policy

> The main data supporting the results in this study are available within the paper and its extended data. The raw datasets generated in this study are available on the Open Data LMU repository at https://data.ub.uni-muenchen.de/713/ in accordance with the principles of open science of the European Research Council (ERC). Whole Genome Sequencing data will be made available for research purposes from the corresponding authors on reasonable request and regulatory clearance.

## Research involving human participants, their data, or biological material

Policy information about studies with human participants or human data. See also policy information about sex, gender (identity/presentation), and sexual orientation and race, ethnicity and racism.

| | |
|---|---|
| Reporting on sex and gender | not applicable |
| Reporting on race, ethnicity, or other socially relevant groupings | not applicable |
| Population characteristics | not applicable |
| Recruitment | Patient tumor samples were collected during surgical procedures at the Dana-Farber/Harvard Cancer Center. |
| Ethics oversight | Patient tumor samples were collected and analyzed according to Dana-Farber/Harvard Cancer Center (DF/HCC) Institutional Review Board (IRB)-approved protocols. |

Note that full information on the approval of the study protocol must also be provided in the manuscript.

# Field-specific reporting

Please select the one below that is the best fit for your research. If you are not sure, read the appropriate sections before making your selection.

☒ Life sciences ☐ Behavioural & social sciences ☐ Ecological, evolutionary & environmental sciences

For a reference copy of the document with all sections, see nature.com/documents/nr-reporting-summary-flat.pdf

# Life sciences study design

All studies must disclose on these points even when the disclosure is negative.

| | |
|---|---|
| Sample size | Samples sizes were chosen based on previously run pilot experiments and are indicated in the figure legends as n values. |
| Data exclusions | All mice used in animal experiments were analyzed except for one mouse (Fig. 5 c-e), which had to be killed due to issues clearly not related to tumor burden or treatment (eye infection). Further, two spleens had to be excluded from Supplementary Figure 1 e Day 9 before data acquisition due to an error in sample preparation leading to cell loss.<br>In Fig. 5 c-e, an outlier test was run and outliers were excluded, as indicated in the figure legend.<br>For in vitro experiments, all technically sound repetitions with appropriate results in the positive and negative controls were included. |
| Replication | For each experiment, the number of independent repeats is clearly stated in the respective figure legends.<br>In vivo experiments have been performed once (Fig. 1 e-g), three time (Fig. 4a-d, Fig. 5a-b) or twice (Fig. 5c-f) , as indicated. Experiments involving patient-dervied tumor samples have been replicated once (Fig. 4g-i) and three times (Fig. 4f). All attempts at replication have been successful.<br>All in vitro experiments have been performed three times using T cells from different donors. All attempts at replication have been successful. |
| Randomization | Mice were stratified into treatment groups by averaging tumor size per group for all animal experiments. |
| Blinding | Researchers were blinded during data acquisition, but not analysis, of all animal experiments. |

# Reporting for specific materials, systems and methods

We require information from authors about some types of materials, experimental systems and methods used in many studies. Here, indicate whether each material, system or method listed is relevant to your study. If you are not sure if a list item applies to your research, read the appropriate section before selecting a response.

## Materials & experimental systems

| n/a | Involved in the study |
|---|---|
| ☐ | ☒ Antibodies |
| ☐ | ☒ Eukaryotic cell lines |
| ☒ | ☐ Palaeontology and archaeology |
| ☐ | ☒ Animals and other organisms |
| ☒ | ☐ Clinical data |
| ☒ | ☐ Dual use research of concern |
| ☒ | ☐ Plants |

## Methods

| n/a | Involved in the study |
|---|---|
| ☒ | ☐ ChIP-seq |
| ☐ | ☒ Flow cytometry |
| ☒ | ☐ MRI-based neuroimaging |

# Antibodies

| Antibodies used | Human Antibodies:<br>Fixable Viability Dye eFluor 780 (1:1000, Invitrogen, Cat.# 65-0865-14), APC anti-human CD3 (1:100, clone OKT3, Biolegend, Cat.# 317318), PE-Cy7 anti-human CD4 (1:100, OKT4, BioLegend, Cat.# 317414), BV605 anti-human CD8 (1:100, SK1, Biolegend, Cat.# 344742), BV421 anti-human CD69 (1:100, clone F50, BioLegend, Cat.#310930), PerCP-Cy5.5 anti-human CD25 (1:100, clone BC96, BioLegend, Cat.# 302626), FITC anti-human/mouse/rat c-myc (1:100, SH1-26E7.1.3, Miltenyi, Cat.# 130-116-485), AF647 anti-CREB (pS133)/ATF-1 (pS63) (1:20, clone J151-21 (RUO), BD Bioscience, Cat.# 558434).<br><br>Murine Antibodies:<br>Pacific Blue anti-CD4 (1:100, clone GK1.5, Biolegend, Cat.#: 100428), FITC anti-CD8 (1:100, clone 53-6.7, Biolegend, Cat.#: 100706), Pacific Blue anti-CD3 (1:100, clone 17A2, Biolegend, Cat.#: 100214), Alexa Fluor 700 anti-CD4 (1:100, clone GK1.5, Biolegend, Cat.#: 100430), BV785 anti-CD8 (1:100, clone 53-6.7, Biolegend, Cat.#: 100750), BV711 anti-CD45.1 (1:100, clone A20, Biolegend, Cat.#: 110739), APC anti-rat CD90/mouse CD90.1 (Thy-1.1) (1:100, clone OX-7, Biolegend, Cat.#: 202526), FITC anti-mouse CD4 (1:100, clone GK1.5, BioLegend, Cat.# 100406), Pacfic Blue anti-mouse CD8a (1:100, clone 53-6.7, BioLegend, Cat.# 100725), PE-Cy7 anti-mouse IFNgamma (1:100, clone XMG1.2, eBioscience, Cat.# 25-7311-82), anti-murine CD3 monoclonal Antibody (145-2C11) Functional Grade eBioscience™ (ThermoFisher Scientific, clone 145-2C11, catalog #16-0031-82), anti-murine CD28 monoclonal Antibody (CD28.2) Functional Grade eBioscience™ (ThermoFisher Scientific, clone 37.51, catalog #16-0281-82) |
|---|---|
| Validation | All antibodies have been validated by the manufacturer and documentation can be found on the respective websites. |

# Eukaryotic cell lines

Policy information about cell lines and Sex and Gender in Research

| Cell line source(s) | Panc02 cells were generated as previously described (Karches and Benmebarek et al. Clin Cancer Res, 2019, Lesch and Blumenberg et al., Nat Biomed Eng, 2021)<br>D4M.3A-OVA cells were generated as previously described (Di Pilato et al., Nature 2019)<br>BxPC3 (from Max Schnurr, Munich, Germany)<br>Msto-hMSLN (from Max Schnurr, Munich, Germany)<br>293Vec-Eco (from Manuel Caruso, Quebec, Canada)<br>293Vec-RD114 (from Manuel Caruso, Quebec, Canada) |
|---|---|
| Authentication | STR DNA profiling of human cell lines. |
| Mycoplasma contamination | Cells were regularly tested for mycoplasma contamination using polymerase chain reaction (PCR). All cells used throughout this study were confirmed to be negative for mycoplasma contamination prior to their use. |
| Commonly misidentified lines (See ICLAC register) | No commonly misidentified cell lines were used in the study. |

# Animals and other research organisms

Policy information about studies involving animals; ARRIVE guidelines recommended for reporting animal research, and Sex and Gender in Research

| Laboratory animals | The following strains of Mus musculus were used:<br>WT mice: C57BL/6J (strain #000664)<br>OT-I mice: C57BL/6-Tg(TcraTcrb)1100Mjb/J (strain #003831)<br>CD45.1 mice: B6.SJL-Ptprca Pepcb/BoyJ (strain #002014)<br>CD90.1 mice: B6.PL-Thy1a/CyJ (strain #000406)<br>Ptger2−/− mice: B6.129-Ptger2tm1Brey/J (strain #004376)<br>Ptger4fl/fl mice : B6.129S6(D2)-Ptger4tm1.1Matb/BreyJ (strain #028102)<br>CD4Cre mice : B6.Cg-Tg(Cd4-cre)1Cwi/BfluJ (strain #022071) |
|---|---|

| | NXG mice: NXG (NOD-Prkdcscid-IL2rgTm1/Rj) |
|---|---|
| | Mice were 6-38 weeks of age. |
| Wild animals | No wild animals were used in the study. |
| Reporting on sex | In all tumor models, sex of the mice was matched to sex of the transplanted tumor cell line (D4M.3A = male; BxPC3 = female, Msto = female). |
| Field-collected samples | No field collected samples were used in the study. |
| Ethics oversight | Local regulatory agency (Regierung von Oberbayern). |

Note that full information on the approval of the study protocol must also be provided in the manuscript.

# Plants

| Seed stocks | not applicable |
|---|---|
| Novel plant genotypes | not applicable |
| Authentication | not applicable |

# Flow Cytometry

## Plots

Confirm that:

☒ The axis labels state the marker and fluorochrome used (e.g. CD4-FITC).

☒ The axis scales are clearly visible. Include numbers along axes only for bottom left plot of group (a 'group' is an analysis of identical markers).

☒ All plots are contour plots with outliers or pseudocolor plots.

☒ A numerical value for number of cells or percentage (with statistics) is provided.

## Methodology

| Sample preparation | Tissue preparation:<br>- mechanical disintegration<br>- only for tumor tissue: collagense/DNase digestion (37°C, 30min)<br>- pass through a cell strainer to get single cell suspensions<br>- erythrocyte lysis if necessary<br>- surface staining (4°C, 30min)<br>- wash step with PBS<br>- FACS analysis in PBS<br><br>Cell culture experiments:<br>- wash T cells with PBS<br>- surface staining (4°C, 30min)<br>- wash with PBS<br>- if applicable: fixation and permeabilization for intracellular stainings<br>- intracellular stainings in Permeabilization Buffer<br>- FACS analysis |
|---|---|
| Instrument | LRSFortessa Cell Analyzer (BD Biosciences)<br>FACS Canto II Flow Cytometer (BD Biosciences)<br>CytoFLEX LX (Beckmann Coulter) |
| Software | BD FACSDiva  (BD Biosciences)<br>CytExpert (Beckmann Coulter)<br>FlowJo (BD Biosciences) |
| Cell population abundance | No cell sorting was performed. |
| Gating strategy | T cell tracking in D4M3A-tumor bearing mice: |

Gating strategy

FSC-A/SSC-A for exclusion of debris and dead cells, FCS-A/FSC-H to select single cells, fixable viability dye to gate on live cells, CD3+ cells to exclude non-T cells, CD45.1/CD90.1 to track respective cell populations

T cell tracking in BxPC3-tumor bearing mice:
FSC-A/SSC-A for exclusion of debris and dead cells, FCS-A/FSC-H to select single cells, fixable viability dye to gate on live cells, CD3+ cells to exclude non-T cells, cMyc and teLuc to track respective cell populations

in vitro CAR T cell experiments:
FSC-A/SSC-A for exclusion of debris and dead cells, FCS-A/FSC-H to select single cells, fixable viability dye to gate on live cells, then depending on the assay further gating on CD4/CD8/EdU/pCREB/IFNg/CD25/CD69

☐ Tick this box to confirm that a figure exemplifying the gating strategy is provided in the Supplementary Information.

