## [Peer Review File · Nature Biomedical Engineering]

Ablation of prostaglandin E2-signaling through dual receptor knock-out in CAR T cells enhances therapeutic efficacy in solid tumors

Corresponding Author: Dr Sebastian Kobold

A version of this paper was originally rejected for publication by Nature Biomedical Engineering, however that decision was reconsidered after appeal by the authors.

Version 0:

Decision Letter:

Dear Sebastian,

Thank you for submitting to *Nature Biomedical Engineering* your Presubmission Enquiry, "Ablation of PGE2-signaling in CAR T cells enhances therapeutic efficacy in solid tumors".

As you know, we screen Presubmission Enquiries against our editorial criteria. These editorial judgements are based on considerations of fit to the journal's scope and, when enough information is provided, of the degree of advance, broad implications, and breadth and depth of the work.

In this case, the topic of the Presubmission Enquiry is of course within the remit of the journal, and we are intrigued by the description of your findings following the knock down of cognate receptors for Prostaglandin E2 after your discovery that it impairs the long-term function of cytotoxic CD8+ T cells. We would thus like to invite you to submit a full manuscript so that we can carry out a full editorial assessment.

I should also ask you to please fill in our [reporting summary](https://www.nature.com/authors/policies/ReportingSummary.pdf) and [policy checklist](https://www.nature.com/authors/policies/Policy.pdf). (Please note that these forms are dynamic PDF files that can only be properly visualized and filled in by using [Acrobat Reader](https://get.adobe.com/reader).)

Both documents are aimed at ensuring good reporting standards and at easing the interpretation of results, and will be available to any reviewers. Should the manuscript be eventually published, the reporting summary will be attached to the published PDF of the paper and will also be available as supplementary information. More information is available on the [editorial policies](http://www.nature.com/authors/policies/availability.html#requirements) page.

Moreover, we highly recommend that you use our [manuscript template](https://bit.ly/3FoGyHd). This will help you ensure that the manuscript complies with our data-presentation recommendations, that it includes all the necessary sections, and that it is structured to facilitate the assessment of peer reviewers and editors. In particular, please make sure that the manuscript provides thorough information on statistics and methods, and that the images comply with our [guidelines for image integrity](https://www.nature.com/nature-portfolio/editorial-policies/image-integrity).

When you are ready to submit the manuscript, please upload the manuscript files as well as the reporting summary and policy checklist. DELETE_IF_COVER_LETTER_ALREADY_PROVIDED Please also upload a brief cover letter that describes the main results of the work and places it into a broader context.

Best wishes,

Pep

Pep Pàmies
Chief Editor, Nature Biomedical Engineering

Version 1:

Decision Letter:

Dear Sebastian,

Thank you again for submitting to *Nature Biomedical Engineering* your manuscript, "Ablation of PGE2-signaling in CAR T cells enhances therapeutic efficacy in solid tumors".

The manuscript has been seen by three experts, whose reports you will find at the end of this message. You will see that although the reviewers appreciate aspects of the work, they raise serious concerns about the overall significance of this study, notably the moderate enhancement of efficacy and the insufficient characterization of the engineered T cells and limited in vivo data. Having considered the reviewers' advice, we have reached the conclusion that even with further work this study is unlikely to reach the scientific significance that we look for in manuscripts that we consider for further external peer review.

We hope that you will find the referee reports helpful when revising the work for submission to a journal with less stringent editorial criteria than *Nature Biomedical Engineering* (we recommend *Nature Communications*, which will allow you to take advantage of the automated transfer of reviewer reports and identities between the journals; please see the footnote below).

Best wishes,

Pep

Pep Pàmies
Chief Editor, Nature Biomedical Engineering

* Although we cannot offer to publish your manuscript, I suggest that you consider Nature Communications as a suitable venue for this work. To transfer your manuscript, please use our manuscript transfer portal. You will not have to re-supply manuscript metadata and files, unless you wish to make modifications. For more information, please see our manuscript transfer FAQ page.

Reviewer #1 (Report for the authors (Required)):

The manuscript by Dorr, Gregor, Lacher and colleagues describes a new CRISPR-based approach to target PGE2 receptors in order to improve the function of CAR T cells. The authors present data with both murine and human CAR T and show that PGE2 limits CAR T proliferation in response to antigen and this leads to improved therapeutic effects in the context of human anti-Her2 CAR T cells. This is associated with significantly increased numbers of CAR T at the tumor site.

The work is the first to describe that CRISPR based targeting of PGE2 receptors can improve the efficacy of CAR T although it should be noted that it is not the first to describe an approach to overcome PGE2 mediated suppression of CAR T (PMID 27045023), nor the first to describe an immunosuppressive role for PGE2 on CAR T (as per reference cited by authors). Although the work further highlights the importance of PGE2 mediated suppression of CAR T cells, its clinical impact is limited due to the modest improvements in efficacy – in the only in vivo CAR T model presented tumor growth is delayed by approximately 15 days and all mice eventually relapse.

Major concerns

1) In terms of CAR T, the study relies solely on one in vivo model in the context of human anti-Her2 CAR T cells. No data for CAR T are provided in a syngeneic model where all immunosuppressive pathways are intact and therefore provides a stringent test for this approach.

2) The murine data are a little confusing. It appears that murine CAR T have been cocultured with human (Panc02) tumor cells. Can this be clarified? Experiments should be performed in a fully murine setting and preferably supported with in vivo data.

3) In both the OT-I and human CAR T model it appears that the improvements in functionality are transient and that the majority of tumors eventually relapse. What is the reason for this, do other cAMP inducing pathways contribute to relapse?

Can responses be further improved by targeting other immunosuppressive pathways e.g. PD-1 simultaneously.

4) The immunological characterization of the mechanism is extremely limited. The authors demonstrate in vitro that proliferation is enhanced following EP2 EP4 double knock out but no changes to cytokine production (no quantification shown) and conclude that the effect is due to a rescue in proliferation. This is surprising as increased cAMP is known to suppress several aspects of T cell activation, including cytokine production. The manuscript should include a thorough assessment of CAR T memory phenotype and effector function in the tumors and spleens of treated mice.

Minor comments

Knock out of EP2 and EP4 is confirmed by functional assessment of cAMP levels following PGE2 administration. Although not critical, given the approach is being developed for clinical application some information on knock out efficiency as well as potential off target editing would further support this.

Reviewer #2 (Report for the authors (Required)):

What is the expression level of EP2 AND EP4 receptors in human and murine (CAR)T cells before knockout? how about patients-derived T cells?

Figure 1a: cAMP is also a downstream molecule other receptors like A2aR, another inhibitory receptor T cells express (PMID: 32151275). Also, it seems that there are some compensatory mechanisms for immune inhibitory receptors where one (or some) is downregulated and the other one(s) upregulated. Have you evaluated the expression of these kinds of receptors? the residual cAMP depicted in Figure 1a might explain this concept.

Why do the authors use different doses of PGE2 for cAMP measurement and testing CAR T cell functionality in vitro?

The major action of PGE2 happens through TME modulation in particular with orchestration with MDSCs and TAMs. These can be evaluated in immunocompetent cancer models.

Reviewer #3 (Report for the authors (Required)):

Based on previous mouse studies in which PGE2 and PGE4 were genetically knocked out in T cells, in this report, the authors use CRISPR to knock out the EP2 and EP4 PGE2 receptors in OT1 cells and in mouse and human CAR T cells. They show that knockout of both of these receptors leads to increased T cell proliferation after exposure to PGE2 in vitro, but, somewhat surprisingly, does not increase T cell activation. Injection of knockout T cells resulted augmented anti-tumor activity in mouse tumor models, again due to increased numbers of T cells rather than more functional T cells.

Overall, this was a well done and interesting paper. Replicate studies and appropriate statistics were used. They raised an interesting issue relating to the main effects of PGE2 being on proliferation versus function. This is a somewhat different result than has been published, but deserves consideration.

The main issues for this reviewer were:

- Insufficient characterization of the results of CRISPR on the T cells
- Apparent treatment of very small tumors (authors need to clarify this)
- Issues of novelty and impact relating to a previous publication from this group- they previously showed effects of loss of PGE2 receptors in adoptive T cell transfer studies. In this paper, they repeat similar studies, but used CAR T cells instead.

Specific Issues:

Major:

L121- is there mRNA and/or protein data showing the actual degree of knockdown of EP2 and EP4, rather than just changes in response to PGE2? How efficient was the CRISPR knockdown for each gene? What percentage of CARTs had EP2 and EP4 knocked down?

Fig. 5. The results showing that the numbers of human CARTs within tumors declines over time is different than most other studies that show an increase in numbers over time with increasing hypofunction occurring. These data were obtained with imaging. Given this discrepancy, can the authors provide any data using direct cell counting (i.e. via flow cytometry) showing this decline in intratumoral CARTs over time?

Minor:

L119- would add "mouse" T cells

L129-please provide the reference here

L157- what was the viability of the final CAR T cell preparations?

Fig 1C- how large were the tumors when CARTs were injected? Would be helpful if the data were in cubic mm.

Fig 4e- the y axis scale should be reduced (to 1.5 at max) to better show the data

Fig 5- how large were the tumors when CARTs were injected?

L256-259- The following statement was confusing to me:

Both in vitro and in vivo, we did not observe a consistently apparent benefit of the EP2 and EP4 double knockout over the individual single knockouts across repetitions and donors. This highlights the necessity of the double knockout to completely abrogate PGE2 signaling and guarantee reliable protection from PGE2 in the TME.

Shouldn't this be: we did not observe a consistently apparent benefit of the EP2 and EP4 single knockouts versus the double knockouts across repetitions and donors.....

L694- Reference 55 is incomplete and in the wrong format- was this paper published in Nature?

Version 2:

Decision Letter:

Dear Dr Kobold,

Please let me begin by apologizing for the length of time your manuscript spent in the second round of review.

I would also like to introduce myself as the new Chief Editor of nBME. I will be handling your manuscript for the final steps.

Thank you for your revised manuscript, "Ablation of prostaglandin E2-signaling through dual receptor knock-out in CAR T cells enhances therapeutic efficacy in solid tumors". Having consulted with reviewer 1, I am pleased to write that we shall be happy to publish the manuscript in *Nature Biomedical Engineering*.

We will be performing detailed checks on your manuscript, and in due course will send you a checklist detailing our editorial and formatting requirements. You will need to follow these instructions before you upload the final manuscript files.

Best wishes,
Rita

Rita Strack, Ph.D.
Chief Editor
Nature Biomedical Engineering

Reviewer #1 (Report for the authors (Required)):

The authors have performed a significant number of experiments that enhance the impact of the study. I do not have any further questions.

Version 3:

Decision Letter:

Dear Sebastian,

I am happy to inform you that your manuscript, "Ablation of prostaglandin E2-signaling through dual receptor knock-out in CAR T cells enhances therapeutic efficacy in solid tumors", has now been accepted for publication in *Nature Biomedical Engineering*.

Over the next few weeks, the figures will be checked for production quality, the text edited to ensure that it conforms to house style, and the manuscript typeset.

Our Articles are published 40-60 days after the acceptance date (we recommend that you inform your institutional press office of this timeframe), and you will be notified of the actual publication date a few days in advance. Articles can be published any working day of the week, and are pushed live shortly after 10 am London time.

Publishing agreement. You will be asked to digitally sign a publishing agreement (grant of rights). After the signed publishing agreement has been received, the proofs of the article will be sent to you for review. If you have any queries during the production process, or you cannot meet the requested deadline for returning the proofs, please contact rjsproduction@springernature.com.

Nature Biomedical Engineering is a Transformative Journal. Authors may publish their research with us through the traditional subscription access route, or make their paper immediately open access through payment of an article-processing charge. More [information about publication options](https://www.springernature.com/gp/open-research/transformative-journals) is available.

You may need to take specific actions to [comply](https://www.springernature.com/gp/open-research/funding/policy-compliance-faqs) with funder and institutional open-access mandates. If the work described in the accepted manuscript is supported by a funder that requires immediate open access (as outlined, for example, by [Plan S](https://www.springernature.com/gp/open-research/plan-s-compliance)) and your manuscript was originally submitted on or after January 1st 2021, then you should select the gold OA route. Authors selecting subscription publication will need to accept our standard licensing terms (including our [self-archiving policies](https://www.springernature.com/gp/open-research/policies/journal-policies)), and these will supersede any other terms that the author or any third party may assert apply to any version of the manuscript.

Acceptance of your manuscript is conditional on agreement, by all authors, with both our [media embargo](http://www.nature.com/authors/policies/embargo.html) and [confidentiality and pre-publicity](http://www.nature.com/authors/policies/confidentiality.html) policies. In particular, you may arrange your own publicity of the Article (for instance, through your institutional press office), as long as you ensure that journalists strictly adhere to the media embargo.

To assist you in disseminating the work, as soon as the Article is published you will be able to take advantage of the Springer Nature [SharedIt](https://www.springernature.com/gp/researchers/sharedit) initiative to [generate a unique shareable link to the Article](http://authors.springernature.com/share) that will allow anyone (with or without a subscription) to read it. Recipients of the link who are subscribers will also be able to download and print the PDF.

Thank you for having submitted this work to *Nature Biomedical Engineering*.

Best wishes,
Rita

Rita Strack, Ph.D.
Chief Editor
Nature Biomedical Engineering

Executive Summary

Here, we would like to briefly address the main concerns regarding our manuscript in a short summary, while you can also find a detailed point-to-point discussion of all reviewers' comments below.

Expert reviewers questioned the novelty of our proposed work, which we can understand in the light of their comments. However, we feel that we did not describe adequately the actual technological advance that turned our attention towards NBME as forum for publication. While CRISPR-Cas9 engineering has been proposed for some years as a strategy to improve therapeutic cells by targeted engineering of for example suppressive molecules, these relied almost exclusively on such surface targets readily detectable at the protein level. Prominent examples include PD-1 but also TIM3¹⁻⁵. Reports on targets not detectable at the protein level for lack of adequate reagents are scarce^{6,7}. Here, we propose for the first time simultaneously targeting two G-protein coupled receptors (GPCR EP2 and EP4) that cannot be directly detected by antibodies, rendering efficiency assessment, especially in a space where protein half-life may survive transcription and translation of the corresponding genes, impossible. To permit such strategy, we capitalized on our knowledge of EP2 and EP4 biology and designed easy to perform functional assays that permit the evaluation of remaining protein function. We thus propose for the first time, that a functional read out can be used to assess editing efficiencies of therapeutic T cells. This advance opens a new target space for therapeutic engineering and is of high value to the field.

To further address the concern of novelty of the approach, we would like to highlight that we do not see the purpose and strength of this manuscript in another study detailing the mechanistic behind PGE₂-mediated T cell inhibition, as we published this in Nature a year ago. Rather, this manuscript is meant as a follow up to provide a clinically relevant engineering solution for this now well described problem. The fact that other publications trying to solve this problem exist, only highlights the need for such a strategy. Our approach to date is the only strategy presented that can prove its efficacy *in vivo*⁸ and in physiologically relevant concentrations of PGE₂⁹, as discussed below (Fig. 1, 4 and 5). Additionally, our strategy functions without the systemic use of small molecule inhibitors or the complete blockade of cAMP signaling in T cells, both of which can hardly be attributed to selective PGE₂-blockade on T cells (as rightly observed by one of the reviewers in his or her comments), and further raise safety concerns, as discussed in the manuscript.

In our revised manuscript we can now provide detailed sequencing information regarding the knockout efficiency of our CRISPR/Cas system, such that we can characterize gene editing efficiencies both on a functional and genomic level. Furthermore, were we able to extensively investigate our EP2 and EP4 knockout strategy in two further models employing anti-HER1 and anti-MSLN CAR T cells, as well as a syngeneic model of anti-EpCAM CAR T cells. To validate our strategy to be clinically relevant, we can provide additional novel data demonstrating the anti-tumor effect of our engineered CAR T cells in patient-derived tumor explant cultures. Lastly, we herein provide experiments in which we investigated the phenotypic state of tumor infiltrating CAR T cells and show increased infiltration and accumulation of EP2 and EP4 knockout CAR T cells into tumors, through our imaging platform, coupled with extensive flow cytometry analysis. Overall, we believe that our revised manuscript entails comprehensive further *in vivo*

characterization adequately supporting our hypothesis, while aligning with the scope of your journal.

Point-to-point discussions of the reviewers' concerns:

Reviewer #1:

"The work is the first to describe that CRISPR based targeting of PGE2 receptors can improve the efficacy of CAR T although it should be noted that it is not the first to describe an approach to overcome PGE2 mediated suppression of CAR T (PMID 27045023), nor the first to describe an immunosuppressive role for PGE2 on CAR T (as per reference cited by authors). Although the work further highlights the importance of PGE2 mediated suppression of CAR T cells, its clinical impact is limited due to the modest improvements in efficacy – in the only *in vivo* CAR T model presented tumor growth is delayed by approximately 15 days and all mice eventually relapse."

We are grateful that this reviewer recognizes the novelty of the engineering approach. We therefore would like to add to his considerations as to why we turned our attention towards NBME as the forum for publication. While CRISPR-Cas9 engineering has been proposed for some years as a strategy to better therapeutic cells by targeted engineering, these primarily relied on surface targets detectable at the protein level to assess knockout efficiency. Prominent examples include PD-1 but also TIM3¹⁻⁵. Here, we propose for the first time simultaneously targeting two G-protein coupled surface receptors (GPCR, EP2 and EP4) that cannot be directly detected by antibodies, rendering efficiency assessment, especially in a space where protein half-life may survive transcription and translation of the corresponding genes, impossible. To permit such strategy, we capitalized on our knowledge of EP2 and EP4 biology and developed easy to perform functional assays which permit the evaluation of remaining protein function. We thus propose for the first time, that a functional read out can be used to assess editing efficiencies of therapeutic T cells. This advance opens a new target space for therapeutic engineering and is of high value to the field.

Next, while we do acknowledge, that impact and targeting of PGE2 on CAR T cells has in principle been proposed before, namely using receptor inhibitors and signaling blockade of the family of downstream effectors^{8,9}, we would like to turn the reviewers' attention to differences that we believe are quite important. Our approach to date is the only strategy presented that can prove its efficacy *in vivo*⁸ and in physiologically relevant concentrations of PGE2⁹. First, quoted publications utilized extreme concentrations of PGE2 that are well above physiological concentrations we found in tumor explants from our xenograft models, as well as from patient-derived tumor samples (discussed below in Fig. 4 and 5). In detail, Akbari et al. only show *in vitro* data with a use of ~3500 ng/ml PGE2. Newick et al. show *in vitro* effects with 5000 ng/ml PGE2 and only limited activity *in vivo*. Importantly, in our hands such extreme concentrations have direct toxic effects on the T cells themselves (Fig. 1, for rebuttal letter only), challenging the validity of these previous findings. In our study, we are able to observe effects with a 10-fold lower amount of PGE2 (500 ng/ml) *in vitro* and robust results *in vivo*. We have now elaborated on these differences in the discussion section of manuscript to permit a better understanding of the advance and relevance of the strategy we are proposing.

Figure 1: Human anti-HER2 CAR T cells were treated with high-dose PGE2 (5000 ng/ml) or vehicle solution. After 48 h the viability was measured by flow cytometry. Cellular viability was reduced by 25 % in the PGE2-treated conditions.

However, this also explains, why no or very limited *in vivo* data was provided in these studies, as such concentrations will not occur and thus effects will not have translated from *in vitro* to *in vivo*. In other words, we reinforce that our study is the first to investigate the physiological and pathophysiological function of PGE2 on CAR T cells with a clear potential for translation.

Along these same lines, the previously proposed PGE2 antagonistic strategies, namely inhibitor celecoxib and complete blockade of cAMP-PKA signaling in T cells come with their own concerns. First, these inhibitors have shown major signs of toxicities in clinical studies as patients experienced extreme adverse events leading to the early abortion of the trial, clearly challenging their suitability for prolonged and sustained application¹⁰⁻¹². Second, pan-inhibition of cAMP and PKA does much more to T cells than pure EP2 or EP4 inhibition. In fact, it plays a key role in immune regulation involving various signaling pathways in T cells, which cannot be reasonably discriminated in the previously discussed paper^{13 14}. In other words, it is speculative, which amount of the previously described effect is due to PGE2 and which to signaling of other agonists and thus cannot preclude novelty towards our own study selective for PGE2. We have further added some wording to our discussion to contrast with this prior art.

In summary, we reason that a) previous studies did in fact not study the impact of PGE2 on CAR T cells but rather induced cell death in these cells, thereby reducing CAR T cell activity unspecifically and b) the proposed approaches for PGE2 antagonism are non-physiologic and come with more safety and efficacy limitations than they solve and are thus unlikely to advance to clinical benefit, highlighting the need for more targeted and efficacious strategies.

This reviewer further rates the clinical potential of our study as limited due to an only modest effect *in vivo*. The presumed limited efficacy of our approach as seen by the relapse is to a great extent caused by our intentional choice of difficult to treat tumor models. The BxPC3 model used for the xenograft experiments in this study is known to be notoriously hard to treat (Mock anti-HER2 CAR T cells in high doses barely show treatment benefits) and was selected because we sought a model that is not adequately responsive to CAR T cells alone to challenge our approach as much as possible. In any case, as can be seen in the dataset below, we can treat this model in a way that induces significant tumor control in mice, as well as inducing transient long term clearance in 3/15 animals when CAR T cells are shielded from PGE2 suppression (Fig. 2 for rebuttal only and Fig. 4 a, b revised manuscript). None of the mice treated with mock electroporated anti-HER2 CAR T cells achieved tumor clearance and in comparison, the survival of mice treated with the double deficient CAR T cells was enhanced by more than two-fold. Furthermore, in effort to provide evidence of broad applicability of our strategy, we now added an additional mesothelioma model using anti-MSLN CAR T cells. Survival of mice was

significantly prolonged and 3 out of 18 mice treated with double-deficient anti-MSLN CAR T cells achieved transient tumor clearance. (Fig. 6 and Fig. 4 c, d revised manuscript). We hope the reviewer will now agree that these substantial additional data sets provide evidence of robust therapeutic benefits, which will need to be confirmed in clinical trials.

Figure 2: NXG mice bearing BxPC3 tumors (10^6 cells s. c.) were treated with 10^7 adoptively transferred (i. v.) anti-HER2 CAR T cells either mock electroporated (green) or gene edited for an EP2 and EP4 knockout (blue). Tumor growth was monitored over time. Data from n=2 biological replicates with n=3-5 mice per group.

Major concerns, Reviewer #1:

1.) In terms of CAR T, the study relies solely on one *in vivo* model in the context of human anti-Her2 CAR T cells. No data for CAR T are provided in a syngeneic model where all immunosuppressive pathways are intact and therefore provides a stringent test for this approach.

We appreciate the reviewers' concern about showing *in vivo* efficacy in only one model. Therefore, we have performed respective experiments in additional xenograft models, such as the MSTO model, as well as probed our CAR T cells against patient-derived tumor samples *ex vivo* (revised manuscript Fig. 4f-i). Please also refer to our elaborations to the previous comment above.

We agree that syngeneic models may better reflect the wealth of suppressive pathways in cancer. First, we would like to highlight that efficacy in a syngeneic model was already demonstrated in the D3M4A-melanoma model through targeting the model antigen Ovalbumin (Fig. 1f, g revised manuscript), thus indicating that the concept is valid in a complex tumor microenvironment and not overshadowed by other suppressive pathways unrelated to PGE2. Nonetheless, to provide additional data in a syngeneic CAR model we now included the following experiment in our supplementary data, which we performed using splenocytes derived from $CD4^{Cre}Ptger2^{-/}Ptger4^{fl/fl}$ mice transduced with the anti-EpCAM-CAR (Fig. 3 of the rebuttal letter, Fig. S1 in the revised manuscript) as a proof of concept for this work. While this data is not based on our CRISPR-model, but relies on a knockout mouse line instead, it still shows that EP2 and EP4-deficient CAR T cells improve functionality, also in a syngeneic tumor model.

Figure 3: C57Bl/6 mice bearing Panc02-OVA-EpCAM tumors (2×10^6 cells s. c.) were treated with 10^7 adoptively transferred (i. v.) anti-EpCAM CAR T cells derived from a **CD4^{Cre}Ptger2^{-/-}Ptger4^{fl/fl}** mouse. Tumor growth and survival were monitored.

Moreover, since establishing this proof-of-concept, we have focused on providing data for our approach in a more clinically relevant human setting. Therefore, we would like to point out that, while PGE2 is produced by immune cells not present in the TME of xenograft models, such as MDSCs for example, such cells are not its only source. The cancer cells itself, as well as cancer associated fibroblasts^{15 16}, which both exist in xenograft models, produce significant levels of PGE2, which we were able to measure in tumor lysates from different xenograft models (Fig. 4 for rebuttal letter only).

Figure 4: NXG mice bearing BxPC3 or Msto-hMSLN tumors (10^6 cells s. c.) were monitored for tumor growth and sacrificed on day 7 post tumor inoculation. Tumor and spleen lysates were submitted to ELISA analysis of PGE2 concentration.

Notably, these amounts of PGE2 are very well in line with the concentrations we were able to measure in pancreatic ductal adenocarcinoma and colorectal carcinoma patient-derived explant cultures in collaboration with Russel Jenkins, MD, PhD at the Massachusetts General Hospital in Boston (Fig. 5d for rebuttal letter only). As these cultures retain the patients microenvironment including MDSCs¹⁷, we believe these cultures to be very indicative of an actual patient setting and therefore used them to study our human CAR T cells.

We took this opportunity to acquire data on testing EP2 and EP4 deficient anti-HER1 CAR T cells on such explant cultures, in which we were also able to demonstrate the benefit of our approach (Fig. 5a-c, revised manuscript Fig. 4c-i). Tumor cell viability in tumor explants significantly dropped when treated with EP2-EP4 knockout anti-HER1 CAR T cells compared to mock CAR or control T cells. This novel data further expands the applicability to a third CAR: anti-HER1 CAR, further underpinning the versatility of EP2-EP4 knock outs to enhance functionality of a wide range of therapeutic receptors: TCR, anti-EpCAM, anti-HER1, anti-HER2 and anti-MSLN CAR T cells.

Figure 5: (A) Pancreatic ductal adenocarcinoma (PDAC), (B) neuroendocrine tumor and (C) colorectal carcinoma (CRC) patient-derived explant cultures were treated with anti-HER1 CAR T cells in a ratio of 1:1 or 1:3 as indicated, either being mock electroporated or lacking both the EP2 and EP4 receptor. (D) Tumor lysates were submitted to ELISA analysis to determine PGE2 concentration. Pooled data from n = 3 independent donors for PDAC and n = 1 donor for CRC, with n = 3 technical replicates.

Xenograft and patient explant models capture highly relevant features for clinical application and, in comparison to syngeneic models, more closely mimic patient conditions due to their human origin. Therefore, we have investigated EP2 and EP4-deficient anti-MSLN CAR T cells in an *in vivo* model employing Msto-hMSLN tumor cells in NXG mice, where we were able to demonstrate efficient tumor reduction and prolonged survival of mice (Fig. 6, revised manuscript Fig. 4c-d).

Figure 6: NXG mice bearing Msto-hMSLN tumors (10^6 cells s. c.) were treated with 2×10^6 adoptively transferred (i. v.) anti-MSLN CAR T cells with EP2 and EP4 knockout or mock electroporated. Mice were monitored for tumor growth and survival over time. Pooled data from $n = 2$ independent experiments, $n = 3 - 5$ mice per treatment group.

2) The murine data are a little confusing. It appears that murine CAR T have been cocultured with human (Panc02) tumor cells. Can this be clarified? Experiments should be performed in a fully murine setting and preferably supported with *in vivo* data.

We apologize if we failed to clearly explain our model here. The Panc02-EpCAM cell line is of murine origin and the mentioned experiments were indeed performed in a fully murine setting as requested, where both the cell line and the antigen (EpCAM) are murine. We now report on *in vivo* CAR T cell data in a syngeneic model (Fig. 3, revised manuscript Fig S1d) using T cells derived from a $CD4^{Cre}Ptger2^{-}/Ptger4^{fl/fl}$ mouse We updated the statement on this experiment in the revised manuscript for clarity, as can be seen in line 152 and 248-249.

3) In both the OT-I and human CAR T model it appears that the improvements in functionality are transient and that the majority of tumors eventually relapse. What is the reason for this, do other cAMP inducing pathways contribute to relapse? Can responses be further improved by targeting other immunosuppressive pathways e.g. PD-1 simultaneously.

As described above, we purposefully chose models that do not react well to CAR T cell therapy to challenge our approach. Nevertheless, the question concerning the reason for relapse is very valid. There might be several possible reasons for the loss of efficacy in our model, not all of which might be related to PGE2. Possible reasons include other cAMP-dependent or –

independent (like PD-1) inhibitory pathways, as well as target antigen loss, which would be entirely independent of the TME. However, it is important to note that many mice, both in the BxPC3 and MSTO tumor model, cleared tumors, indicating that the approach can be sufficiently potent as stand alone, while unrelated factors may drive resistance or relapse and may need concurrent targeting as previously suggested by us and others previously. One challenge to the further proposed experiments is the feasibility of finding a window for efficacious single treatments, such as our approach or the suggested PD1-blockade would synergize. Based on our previous experience this would require extensive engineering and model variations, which would be clearly out of the scope of a study targeted to EP2-EP4 specifically. We thus respectfully reason that the additional data now provided in mice and human models, resulting in clearance in a substantial proportion of settings adequately reflects relevant therapeutic efficiency.

4) The immunological characterization of the mechanism is extremely limited. The authors demonstrate *in vitro* that proliferation is enhanced following EP2 EP4 double knock out but no changes to cytokine production (no quantification shown) and conclude that the effect is due to a rescue in proliferation. This is surprising is increased cAMP is known to suppress several aspects of T cell activation, including cytokine production. The manuscript should include a thorough assessment of CAR T memory phenotype and effector function in the tumors and spleens of treated mice.

As mentioned above, we see the main focus and value of this manuscript in providing a clinically feasible strategy to counter the negative effect of PGE2 in CAR T cells. We have recently published a very comprehensive mechanistic study detailing the effect of PGE2 on T cells describing that the influence of PGE2 on T cells is mainly carried out by downregulation of IL-2 signaling leading to defects in proliferation and not activation. As we already published mechanistic data, we wanted to keep the focus of this manuscript on the engineering and translational aspects. The reviewer mentions the lack of cytokine production, which we have addressed by an intracellular staining for IFN γ production in the revised manuscript Fig. 3, where we could not see any differences caused by PGE2.

Further, the reviewer asks for a thorough assessment of therapeutic T cell differentiation and function *in vivo*. We have performed *in vivo* phenotyping experiments in a murine model, whereby we sought to phenotype therapeutic T cells in the spleen and tumor of treated mice. This data was acquired in the tracking experiment is presented in Fig. 1a-c of our revised manuscript, where we could show that wild type OT-I cells fail to accumulate in tumors. While we did acquire data regarding the differentiation and activation phenotype of the T cells in this experiment additionally to cell numbers provided in the revised manuscript, we did not include those as the cell numbers of the wild type cells in the tumor were too low to allow a reliable comparative analysis of their phenotype. This again underlines the mechanism we postulate above, whereby the exposure of T cells to PGE2 suppresses proliferation. However, in the spleens of those mice, we were able to acquire the required phenotyping data where we could detect sufficient amounts of T cells for both EP2^{-/-}EP4^{-/-}, as well as mock T cells. Here, we could not observe any differences in their phenotypic state in terms of differentiation (Fig. 8, for rebuttal letter only).

Figure 7: 5×10^6 CD45.1 x CD4^{Cre}Ptger2^{-/-}Ptger4^{fl/fl} OT-I T cells and 5×10^6 CD90.1 OT-I T cells were co-injected into D4M.3A-SIINFEKL (10^6 cells s. c.) bearing mice. On days 2, 5, 9 and 14 the differentiation state of T cell populations in spleens was analyzed by flow cytometry.

Furthermore, we have also investigated the phenotypic state of anti-HER2 CAR T cells in the previously described human NXG model. We terminated mice 12 days after treatment with anti-HER2 CAR T cells and performed flow cytometric analysis of tumor infiltrating CAR T cells. We saw an increased infiltration of tumors with EP2 and EP4 deficient anti-HER2 CAR T cells in comparison to mock electroporated cells, both in terms of CD8⁺ T cells and cells positive for the telLuc-mCherry receptor (Fig. 9, revised manuscript Fig. 5c). This can be explained by the increased expansion of therapeutic CAR T cells once shielded from the PGE2 in the TME.

Figure 8: NXG mice bearing BxPC3 tumors (10^6 cells s. c.) were treated with 10^7 adoptively transferred (i. v.) anti-HER2 CAR T cells with EP2 and EP4 knockout. Experiment was terminated 12 days after CAR T cell treatment and tumors were analyzed using flow cytometry. Relative count of CD8⁺ and telLuc⁺ cells within the viable cell population depicted.

Interestingly, no apparent differences were observed when phenotyping CAR T cells, neither in terms of their activation or exhaustion status, nor in terms of differentiation. The exhaustive state of the anti-HER2 CAR T cells, in terms of PD-1, TIM-3 and LAG-3, showed no relevant differences between the treatment groups, while CD69 expression also seemed to be the same between the different groups (Fig. 10, revised manuscript Fig. 5d-f). As demonstrated in our first publication by Lacher et al. published in Nature, and our present *in vitro* phenotyping of activated anti-HER2 CAR T cells (revised manuscript Fig. 3i-j), we hypothesize that the benefit in anti-

tumor effect is due to increased expansion, rather than in changes of the phenotypic status. The data presented here aligns with previous data, as we see an increased number of therapeutic CAR T cells mediating the anti-tumor effect, rather than changes in activation or differentiation.

Figure 9: NXG mice bearing BxPC3 tumors (10^6 cells s. c.) were treated with 10^7 adoptively transferred (i. v.) anti-HER2 CAR T cells with EP2 and EP4 knockout. Experiment was terminated 12 days after CAR T cell treatment and tumors were analyzed using flow cytometry. Relative count of positive cells from CD8⁺ T cells depicted. Pooled data from n = 2 independent experiments with n = 4-5 mice per treatment group shown.

Minor comments:

Knock out of EP2 and EP4 is confirmed by functional assessment of cAMP levels following PGE2 administration. Although not critical, given the approach is being developed for clinical application some information on knock out efficiency as well as potential off target editing would further support this.

We fully agree that a more in-depth characterization of the knockout CAR T cell product is necessary for a clinical translation, such that we have performed targeted next generation sequencing to quantify the knockout efficiency of our gRNAs. In this experiment, all gRNA target sites are amplified by a PCR, adaptors are added to each sample for multiplexing and the resulting PCR products are pooled and sequenced in an Illumina MiSeq sequencing system. This gave us a full profile of all sequences at the gRNA target sites, so we were able to evaluate the level of knockout efficiency for each separate gRNA, as well as quantify the proportion of sequences leading to a full knockout for both EP2 and EP4 separately as well as together (Fig. 11, revised manuscript Fig. 2d).

For EP2, in three human donors transduced with the anti-HER2 CAR we achieved a knockout efficiency of nearly 60 % with the first gRNA and nearly 40 % with the second gRNA (Fig. 11, revised manuscript Fig. 2d). As both gRNAs were used simultaneously, this leads to a combined knockout efficiency of around 70 % for EP2. For EP4, gRNA 1 achieved a knockout efficiency of around 95 % and gRNA 2 achieved a 90 % knockout. Together, an estimated knockout efficiency of more than 90 % can be expected. Notably, knockout efficiencies of both EP2 and EP4 were similarly high when combined in a double receptor knockout setting, indicating that knockouts are compatible with each other and a simultaneous knockout is feasible. In summary, we can

substantiate high knockout efficiencies of around 70 % for EP2 and around 95 % for EP4 in three human donors.

Figure 10: Targeted next generation sequencing of anti-HER2 CAR T cells lacking either the EP2 or EP4 receptor or both simultaneously. All gDNA target sites were amplified by a PCR, adaptors are added to each sample for multiplexing and the resulting PCR products are pooled and sequenced in an Illumina MiSeq sequencing system. Percentage of in-frame and out-of-frame indels is quantified, indicating the knockout efficiency of gRNA 1 and gRNA 2. Data from three independent donors, n = 3.

We also appreciate the reviewers comment regarding potential off-target editing. Therefore, we have included an off-target analysis into Fig. 2 of the revised manuscript. Briefly, we performed whole genome sequencing (WGS) of single and double knockouts, as well as mock and untransduced samples from one human donor. We predicted off-targets for each gRNA with the Cas-OFFinder tool allowing up to 4 mismatches between gRNA and DNA sequences (Fig. 12 and Fig 2e of the revised manuscript). We manually checked all predicted off-target site in our WGS data using the integrative genomics viewer (IGV) and could not confirm any off the predicted off-targets.

Figure 11: Number of potential off-targets predicted by Cas-OFFinder allowing up to 4 mismatches for each gRNA.

Additionally, for an unbiased approach, we performed variant calling (GATK) of our samples in comparison to the reference genome hg38. We excluded variants also present in the mock and untransduced control samples as well as variants that were not reproducible between replicates. The remaining variants were filtered for their likelihood to cause changes on protein level (high priority) (Fig. 13 and Fig. 2f of the revised manuscript). The remaining variants were manually checked in the IGV additionally. The only variants that could be confirmed where the target genes PTGER2 and PTGER4 in their respective knockout samples, proving the ability of our analysis to find actual editing events. Overall, while only performed in one donor, we could not find any evidence of off-target editing for our approach.

Figure 12: Workflow for filtering WGS data for relevant variants. First, our raw WGS data was mapped to the reference genome hg38 and variant calling was performed using the GATK algorithm. All variants present in the control samples (Mock-CRISPRed and untransduced (UT)) were excluded from single and double knockout samples, as they are likely unrelated to the CRISPR-editing and represent normal variations of our donor. Next, variants that could not be reproducibly observed between the single and double knockouts were excluded. Lastly, from the remaining variants we filtered high priority variants likely leading to alteration on protein level.

Reviewer #2:

What is the expression level of EP2 AND EP4 receptors in human and murine (CAR)T cells before knockout? how about patients-derived T cells?

As detailed before, we have not been able to identify specific antibodies to determine the protein levels of EP2 and EP4 by flow cytometry, however previous literature has described the presence of both EP2 and EP4 on T cells^{18,19}. However, to provide a further estimate of expression beyond functional read outs, we analyzed mRNA expression of the target genes in T cells from three human donors via qPCR (Fig. 13, rebuttal letter only). Here we could clearly observe expression of EP2 and EP4, whereby EP2 expression seems to be higher than EP4 expression in T cells derived from health donors.

Figure 13: Relative mRNA expression of EP2 and EP4 genes in wildtype T cells derived from healthy donors. Relative expression determined through normalization to β -actin gene expression. Data from three independent donors, n = 3.

Figure 1a: cAMP is also a downstream molecule of other receptors like A2aR, another inhibitory receptor T cells express (PMID: 32151275). Also, it seems that there are some compensatory mechanisms for immune inhibitory receptors where one (or some) is downregulated and the other one(s) upregulated. Have you evaluated the expression of these kinds of receptors? The residual cAMP depicted in Figure 1a might explain this concept.

We fear there may be a misunderstanding in the interpretation of our previous data: In the cAMP assay depicted in our original manuscript Fig. 1, now revised manuscript Fig. 2, cells are stimulated with purified PGE2 inducing cAMP under forced conditions designed to show blocked PGE2 signaling as a measurement for a successful knockout. No other cAMP stimulants like adenosine were added. As the calculation of cAMP levels entails a subtraction of background signal measured on unstimulated T cells, all cAMP inducing signals unrelated to PGE2 would have been excluded from the calculation. While other cAMP-inducing pathways like adenosine exist and are of relevance (as we have also published previously²⁰), they are not pertinent for the interpretation of this assay.

Regardless, we are happy to further consider the possibility of the compensatory upregulation of other cAMP-inducing receptors like A2aR. Therefore, we investigated the effect of the EP2 and EP4 knockout on adenosine signaling. However, based on our previously published work on this receptor in CAR T cells, we measured unabated signaling caused by the stable adenosine analogue NECA on EP2^{-/-}EP4^{-/-} compared to mock CRISPRed CAR T cells. This can adequately demonstrate regulation of adenosine signaling or not. As shown in Fig. 14 (for rebuttal letter only), while cAMP is not induced by PGE2 on EP2^{-/-}EP4^{-/-} CAR T cells, cAMP amounts increase slightly when T cells are treated with NECA. Further, the amount of cAMP induced by NECA is comparable between EP2^{-/-}EP4^{-/-} and mock electroporated CAR T cells, indicating that knocking

out PGE2 receptors EP2 and EP4 does not lead to a compensatory upregulation of adenosine-induced signaling.

Figure 14: Induction of cAMP by PGE2 and the adenosine-analogue NECA in murine mock and EP2^{-/-}EP4^{-/-} anti-EpCAM-CAR T cells. Data of n = 6 biological replicates.

Why do the authors use different doses of PGE2 for cAMP measurement and testing CAR T cell functionality in vitro?

Different amounts of PGE2 were used throughout the manuscript to address different needs of the respective assays. We have chosen higher amounts of PGE2 to show blockade of PGE2 signaling as a proxy for knockout efficiency to demonstrate high efficiencies in a forced setting with unusually high amounts of PGE2. We previously performed a titration of PGE2 concentration (30-2000 ng/ml PGE2) for the cAMP assay and high doses of PGE2 were chosen to show robustness of the knockouts. For functional assays, we chose much lower concentrations of PGE2 to show the benefit of our approach at even lower and more physiological concentrations of PGE2, while avoiding toxic effects on T cells (as shown above in Fig. 1 for rebuttal letter only). This was further clarified in our manuscript in line 200-203.

The major action of PGE2 happens through TME modulation in particular with orchestration with MDSCs and TAMs. These can be evaluated in immunocompetent cancer models.

We agree on the context dependent action of PGE2 and as described above, MDSCs and TAMs are not the only sources of PGE2 and we were able to detect PGE2 production also in tumor lysates from xenograft models (Fig. 4, for rebuttal letter only). Further, we could demonstrate efficacy in two syngeneic models using OT-I and anti-murine EpCAM CAR T cells with knockout of EP2 and EP4 (Fig. 3, revised manuscript Fig. 1 f-g and Fig. S1d). Together this data indicates our strategy to be of value in immunocompetent cancer models.

Additionally, we could now demonstrate anti-tumor efficacy of our strategy in a patient explant scenario containing an actual patient TME. We have treated pancreatic ductal adenocarcinoma, as well as colorectal carcinoma tumor explants with anti-HER1 CAR T cells and have sufficient decrease of viability when treating with EP2 and EP4 deficient CAR T cells, both in comparison to wildtype T cells and mock electroporated anti-HER1 CAR T cells (Fig. 5, revised manuscript

Fig. 4f-i). Together, these experiments address the raised concerns from different facets both in murine and human settings.

Reviewer #3:

The main issues for this reviewer were:

- Insufficient characterization of the results of CRISPR on the T cells

We again agree with the importance of characterizing the impact of CRISPR knock out on our CAR T cells. As described above to reviewer #2, we have already determined the knockout efficiencies for both receptors by sequencing and could achieve high efficiencies of 70 % for EP2 and 95 % for EP4 (Fig. 11, revised manuscript Fig. 2d). Please refer to that comment for further detail.

- Apparent treatment of very small tumors (authors need to clarify this)

All mice were treated when tumors were established. By definition this means, that we only included tumors, where spontaneous rejections or rejections by control treatments are excluded^{21 22}. By nature, this is highly dependent on the model. As mentioned, the initial model (BxPC3) is a model that is very aggressive and inherently resistant to CAR T cells (no or little impact of CAR T cells alone). In any case, we could generate novel data using an optimizing treatment regimen, in which our PGE2-resistant CAR T cells can even lead to tumor rejections in this model (Fig. 2, revised manuscript Fig. 4a-b). To further address the concern that results may be inflated by small tumor sizes at treatment, we now conducted further experiments, where tumors (Msto) were grown to 40 mm² and treated with a single infusion of anti-mesothelin CAR T cells (with or without EP2-EP4 KO), demonstrating strong anti-tumoral efficacy as well (Fig. 6, revised manuscript Fig. 4c-d). Together our novel data indicate that our strategy leads to substantial efficacy in a range of cancer models, independent of the size of tumor at treatment onset.

- Issues of novelty and impact relating to a previous publication from this group- they previously showed effects of loss of PGE2 receptors in adoptive T cell transfer studies. In this paper, they repeat similar studies, but used CAR T cells instead.

In contrast to reviewer 1, reviewer 3 seems to worry about the similarity to our mechanistic study recently published in Nature. We would like to draw your attention to the fact that the focus of our previous publication was to provide a fundamental mechanistic explanation as to why and how PGE2 negatively affects T cells, thereby presenting itself as a fundamental biological research study. Importantly as well, our published manuscript studied the impact of PGE2 on stem or stem-like T cells as naturally occurring orchestrators of tumor immunity²³. These cells are absent from a therapeutically engineered cell population, which is a result of the isolation and culture process required thereto²⁴. This means, that the cellular product that we are using, and that would be used clinically to treat patients, has no phenotypical overlap with the cell populations we previously studied, therefore it was unclear how this would apply to more differentiated T cell populations. In this sense, the present manuscript submitted for consideration expands our knowledge by demonstrating a similar impact on differentiated effector and therapeutic cells. As proof of this major difference in cellular populations, Lacher

et al. employed as little as 10^3 OT1 T cells for transfer experiments, whereby in our therapeutic setting doses ranging from 10^6 - 10^7 were required for an anti-tumor effect. This demonstrates the fundamental difference in nature of these cells.

As elaborated above, the focus of this manuscript, in contrast to the previous publication, is not a characterization of the effect of PGE2 on CAR T cells but on the first-in class engineering strategy capitalizing on basic biology knowledge. We therefore consider this manuscript to be the logical follow up to our previous work that provides the key engineering solution that our first manuscript, as well as other publications, do not offer.

We would like to further underpin the engineering complexity of our strategy itself, which we did not sufficiently address in the initial version of this manuscript, but have now elaborated upon. GPCRs are notoriously difficult targets to stain, to which EP2 and EP4 are no exceptions. As mentioned in our submitted manuscript, we were not able to identify any antibodies that allow for a reproducible and specific staining of both receptors. Consequently, developing a successful double receptor knockout including the choice of efficient guide RNAs and optimizing the CRISPR protocol in accordance with the CAR transduction protocol were highly challenging. With this manuscript, however, we can demonstrate for the first time a workflow to target challenging, non-stainable targets in a double knockout system with high efficiency in primary human CAR T cells. We have clarified this further in our revised manuscript in lines 176-180 and 402-408.

L121- is there mRNA and/or protein data showing the actual degree of knockdown of EP2 and EP4, rather than just changes in response to PGE2? How efficient was the CRISPR knockdown for each gene? What percentage of CARTs had EP2 and EP4 knocked down?

We anticipated this question, such that the missing data requested has been described to reviewer #2 previously and is included in the revised manuscript Fig. 2. In brief, as described above, we conducted targeted next generation sequencing of genomic DNA of three human donors at the CRISPR target site for EP2 and EP4. From this sequencing data, we estimate that on average we reach 70 % and 90 % knockout efficiency for EP2 and EP4, respectively (Fig. 11 revised manuscript Fig. 2). Furthermore, we have performed whole genome sequencing from one donor but did not find significant off-target editing events, indicating the genome-related safety of our approach (Fig. 12 und 13, revised manuscript Fig. 2e-f).

Fig. 5. The results showing that the numbers of human CARTs within tumors declines over time is different than most other studies that show an increase in numbers over time with increasing hypofunction occurring. These data were obtained with imaging. Given this discrepancy, can the authors provide any data using direct cell counting (i.e. via flow cytometry) showing this decline in intratumoral CARTs over time?

We want to highlight that the data presented in the originally submitted manuscript, as well as in the revised manuscript Fig. 5, shows the luminescence measured normalized to the tumor size on the according day of measurement. Therefore, while smaller tumors demonstrated large signals and thus an increased expansion of therapeutic T cells, the luminescence signal remained rather steady over time, even while tumors relapsed. To further clarify this, we here also present the raw average radiance obtained over the experimental period (Fig. 15, for

rebuttal letter only). While in the beginning we were able to observe an expansion of therapeutic T cells, the signal remained in a rather steady range over a 40-day experimental time frame. Additionally, we did observe an expansion of therapeutic CAR T cells when performing flow cytometry-based single cell analysis of our treated mice within the first 12 days of the treatment time frame (Fig. 9, revised manuscript Fig. 5c).

Figure 15: NXG mice bearing BxPC3 tumors (10^6 cells s. c.) were treated with 10^7 adoptively transferred (i. v.) anti-HER2 CAR T cells with EP2 and EP4 knockout. Tumor growth was monitored over time.

Minor Comments:

L119- would add "mouse" T cells

The type of T cells used, human or mouse, has been clarified throughout the revised manuscript.

L157- what was the viability of the final CAR T cell preparations?

From our isolation protocols we generate CAR T cells with a viability of approximately 80 % for murine and human cells, both gated from the initial lymphocyte gate as shown below (Fig. 16).

Figure 16: Representative gating strategy of murine lymphocytes. Specific gating indicated in each plot.

Fig 1C- how large were the tumors when CARTs were injected? Would be helpful if the data were in cubic mm.

All mice were treated when tumors were established. By definition this means, that we only included tumors, where spontaneous rejections or rejections by control treatments are excluded^{21 22}. Thus, tumors were grown to a size of about 40 mm² before initiating treatment with the therapeutic T cells. For our purposes, we present tumor size as mm², as we believe that this presents a more accurate tumor size, compared to mm³, which by nature imply mathematical extrapolation of the third measure dimension (width and height can be measured not thickness of the tumor).

Fig 4e- the y axis scale should be reduced (to 1.5 at max) to better show the data

We have updated our figures and thus also applied suggested changes by the reviewer (revised manuscript Fig. 3).

Fig 5- how large were the tumors when CARTs were injected?

All mice were treated when tumors were established. By definition this means, that we only included tumors, where spontaneous rejections or rejections by control treatments are excluded^{21 22}. Thus, tumors were grown to a size of about 40 mm² before initiating treatment with the therapeutic T cells.

L256-259- The following statement was confusing to me: *Both in vitro and in vivo, we did not observe a consistently apparent benefit of the EP2 and EP4 double knockout over the individual single knockouts across repetitions and donors. This highlights the necessity of the double knockout to completely abrogate PGE2 signaling and guarantee reliable protection from PGE2 in*

the TME. Shouldn't this be: we did not observe a consistently apparent benefit of the EP2 and EP4 single knockouts versus the double knockouts across repetitions and donors....

We agree with reviewer #3 that the initial passage found in the discussion of our originally submitted manuscript is unclear and have thus updated the text as suggested in our revised manuscript (lines 361-363).

L694- Reference 55 is incomplete and in the wrong format- was this paper published in Nature?

We have updated the format of reference number 55 in our revised manuscript. The reference refers to Lacher and Dörr et al. 2024, which is our previously published manuscript in Nature. The correct reference can now be found as reference #55 in our revised manuscript.

References

1. Gao Q, Dong X, Xu Q, et al. Therapeutic potential of CRISPR/Cas9 gene editing in engineered T-cell therapy. *Cancer Medicine* 2019;8(9):4254-64. doi: <https://doi.org/10.1002/cam4.2257>
2. Li C, Mei H, Hu Y. Applications and explorations of CRISPR/Cas9 in CAR T-cell therapy. *Briefings in Functional Genomics* 2020;19(3):175-82. doi: 10.1093/bfgp/elz042
3. Lin Y, Chen S, Zhong S, et al. 350 - Phase I clinical trial of PD-1 knockout anti-MUC1 CAR-T cells in the treatment of patients with non-small cell lung cancer. *Annals of Oncology* 2019;30:xi12. doi: <https://doi.org/10.1093/annonc/mdz448>
4. Lu Y, Xue J, Deng T, et al. A phase I trial of PD-1 deficient engineered T cells with CRISPR/Cas9 in patients with advanced non-small cell lung cancer. *Journal of Clinical Oncology* 2018;36(15_suppl):3050-50. doi: 10.1200/JCO.2018.36.15_suppl.3050
5. Stadtmauer EA, Fraietta JA, Davis MM, et al. CRISPR-engineered T cells in patients with refractory cancer. *Science* 2020;367(6481):eaba7365. doi: doi:10.1126/science.aba7365
6. Dimitri A, Herbst F, Fraietta JA. Engineering the next-generation of CAR T-cells with CRISPR-Cas9 gene editing. *Molecular Cancer* 2022;21(1):78. doi: 10.1186/s12943-022-01559-z
7. Insel PA, Sriram K, Wiley SZ, et al. GPCRomics: GPCR Expression in Cancer Cells and Tumors Identifies New, Potential Biomarkers and Therapeutic Targets. *Frontiers in Pharmacology* 2018;9
8. Akbari B, Soltantoyeh T, Shahosseini Z, et al. PGE2-EP2/EP4 signaling elicits mesoCAR T cell immunosuppression in pancreatic cancer. *Front Immunol* 2023;14:1209572. doi: 10.3389/fimmu.2023.1209572 [published Online First: 20230630]
9. Newick K, O'Brien S, Sun J, et al. Augmentation of CAR T-cell Trafficking and Antitumor Efficacy by Blocking Protein Kinase A Localization. *Cancer Immunology Research* 2016;4(6):541-51. doi: 10.1158/2326-6066.Cir-15-0263
10. Jakobsen A, Mortensen JP, Bisgaard C, et al. A COX-2 inhibitor combined with chemoradiation of locally advanced rectal cancer: a phase II trial. *Int J Colorectal Dis* 2008;23(3):251-5. doi: 10.1007/s00384-007-0407-7 [published Online First: 20071207]
11. Reyners AKL, de Munck L, Erdkamp FLG, et al. A randomized phase II study investigating the addition of the specific COX-2 inhibitor celecoxib to docetaxel plus carboplatin as first-line chemotherapy for stage IC to IV epithelial ovarian cancer, Fallopian tube or primary peritoneal carcinomas: the DoCaCel study. *Ann Oncol* 2012;23(11):2896-902. doi: 10.1093/annonc/mds107 [published Online First: 20120611]
12. Gitlitz BJ, Bernstein E, Santos ES, et al. A Randomized, Placebo-Controlled, Multicenter, Biomarker-Selected, Phase 2 Study of Apricoxib in Combination with Erlotinib in Patients with Advanced Non-Small-Cell Lung Cancer. *Journal of Thoracic Oncology* 2014;9(4):577-82. doi: <https://doi.org/10.1097/JTO.000000000000082>
13. Wehbi VL, Taskén K. Molecular Mechanisms for cAMP-Mediated Immunoregulation in T cells - Role of Anchored Protein Kinase A Signaling Units. *Front Immunol* 2016;7:222. doi: 10.3389/fimmu.2016.00222 [published Online First: 20160608]
14. Mosenden R, Taskén K. Cyclic AMP-mediated immune regulation--overview of mechanisms of action in T cells. *Cell Signal* 2011;23(6):1009-16. doi: 10.1016/j.cellsig.2010.11.018 [published Online First: 20101202]
15. Mizuno R, Kawada K, Sakai Y. Prostaglandin E2/EP Signaling in the Tumor Microenvironment of Colorectal Cancer. *International Journal of Molecular Sciences* 2019;20(24):6254.
16. Xiang X, Niu Y-R, Wang Z-H, et al. Cancer-associated fibroblasts: Vital suppressors of the immune response in the tumor microenvironment. *Cytokine & Growth Factor Reviews* 2022;67:35-48. doi: <https://doi.org/10.1016/j.cytogfr.2022.07.006>
17. Sun Y, Maggs L, Panda A, et al. TBK1 Targeting Is Identified as a Therapeutic Strategy to Enhance CAR T-Cell Efficacy Using Patient-Derived Organotypic Tumor Spheroids. *Cancer Immunology Research* 2025;13(2):210-28. doi: 10.1158/2326-6066.Cir-23-1011
18. Punyawatthanakool S, Matsuura R, Wongchang T, et al. Prostaglandin E2-EP2/EP4 signaling induces immunosuppression in human cancer by impairing bioenergetics and ribosome biogenesis in immune cells. *Nature Communications* 2024;15(1):9464. doi: 10.1038/s41467-024-53706-3
19. Wang J, Zhang L, Kang D, et al. Activation of PGE2/EP2 and PGE2/EP4 signaling pathways positively regulate the level of PD-1 in infiltrating CD8(+) T cells in patients with lung cancer. *Oncol Lett* 2018;15(1):552-58. doi: 10.3892/ol.2017.7279 [published Online First: 20171026]

20. Seifert M, Benmebarek MR, Briukhovetska D, et al. Impact of the selective A2(A)R and A2(B)R dual antagonist AB928/etrumadenant on CAR T cell function. *Br J Cancer* 2022;127(12):2175-85. doi: 10.1038/s41416-022-02013-z [published Online First: 20221020]
21. Tan M, Fang H-B, Tian G-L. Statistical Analysis for Tumor Xenograft Experiments in Drug Development. *Contemporary Multivariate Analysis and Design of Experiments*:351-68.
22. Zavrakidis I, Józwiak K, Hauptmann M. Statistical analysis of longitudinal data on tumour growth in mice experiments. *Scientific Reports* 2020;10
23. Sebastian B, Lacher JD, Gustavo P. de Almeida, Julian Hönninger,, Felix Bayerl AH, Anna-Marie Pedde, Philippa Meiser, Lukas Ramsauer, Thomas J. Rudolph, Nadine Spranger, Matteo Morotti, Alizee J. Grimm,, Sebastian Jarosch AO, Lisa Gregor, Stefanie Lesch, Stefanos Michaelides, Luisa Fertig, Daria Briukhovetska, Lina Majed, Sophia Stock, Dirk H. Busch,, et al. PGE2 limits effector expansion of + tumour-infiltratingstem-likeCD8 T cells. *Nature* 2024
24. Stoiber S, Cadilha BL, Benmebarek MR, et al. Limitations in the Design of Chimeric Antigen Receptors for Cancer Therapy. *Cells* 2019;8(5) doi: 10.3390/cells8050472 [published Online First: 20190517]

What is the expression level of EP2 AND EP4 receptors in human and murine (CAR)T cells before knockout? how about patients-derived T cells?

Figure 1a: cAMP is also a downstream molecule other receptors like A2aR, another inhibitory receptor T cells express (PMID: 32151275). Also, it seems that there are some compensatory mechanisms for immune inhibitory receptors where one (or some) is downregulated and the other one(s) upregulated. Have you evaluated the expression of these kinds of receptors? the residual cAMP depicted in Figure 1a might explain this concept.

Why do the authors use different doses of PGE2 for cAMP measurement and testing CAR T cell functionality in vitro?

The major action of PGE2 happens through TME modulation in particular with orchestration with MDSCs and TAMs. These can be evaluated in immunocompetent cancer models.